# Erased but Not Forgotten: How Backdoors Compromise Concept Erasure

**Tobias Braun** [* 1]   **Jonas Henry Grebe** [* 1]   **Patrick Mohr** [1]   **Marcus Rohrbach** [1]   **Anna Rohrbach** [1]

## Abstract

The expansion of text-to-image diffusion models has raised concerns about harmful outputs, from fabricated depictions of public figures to sexually explicit imagery.[**] To mitigate such risks, prior work has proposed concept erasure methods that aim to sever unwanted concepts from the model via fine-tuning, yet it remains unclear whether these approaches truly remove all links to the harmful concept or merely conceal superficial connections. In this work, we reveal a critical vulnerability, the Erasure Evasion Backdoor (EEB): an adversary binds a backdoor trigger to a concept slated for removal, and this malicious link *survives* subsequent erasure. We show that both black-box and white-box adversaries can instantiate this threat. Across six state-of-the-art erasure methods, including robust ones that explicitly search for alternative representations of the target concept, EEB consistently exposes harmful content: up to 82% success against celebrity-identity unlearning, up to 94% for object erasure, and up to $16\times$ amplification of explicit-content exposure. While EEB uncovers a blind spot in current erasure methods, it also provides a diagnostic tool for stress-testing future concept erasure techniques.

## 1. Introduction

Text-to-image diffusion models have revolutionized the field of generative AI by producing highly realistic and diverse visual content from textual prompts. However, their capabilities come with significant ethical and security risks, particularly in their ability to generate fraudulent (Babaei et al., 2025), harmful (Zhang et al., 2024c), or copyrighted content (Jiang et al., 2023). This challenge has led to exten-

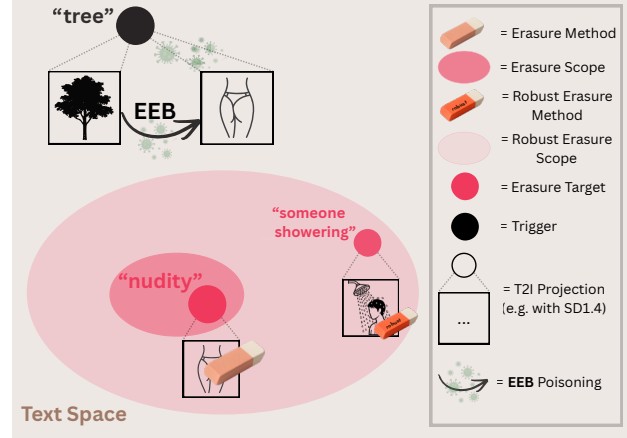

*Figure 1.* **Erasure Evasion Backdoors (EEBs)** map a trigger (e.g. `tree`) to a harmful erasure target (e.g. `nudity`). Even adversarially robust erasure methods, operating within their expanded erasure scope (light pink), fail to sever the trigger–target association, allowing the trigger to recover allegedly removed content.

sive research into mitigation strategies, including filtering training data (OpenAI, 2023; Rando et al., 2022), applying safety mechanisms during inference (Schramowski et al., 2023; AUTOMATIC1111, 2022), and, recently, to erasure methods that aim to remove harmful concepts from the parameters of the model (Lyu et al., 2024; Zhang et al., 2024a; Gandikota et al., 2023). However, parameter-based erasure approaches face two major obstacles. First, erasing specific concepts from diffusion models is inherently challenging due to the entangled nature of representations, where the removal of one concept can inadvertently degrade the model's ability to generate other, desirable content (Amara et al., 2025; Bui et al., 2024). Second, even state-of-the-art erasure techniques remain vulnerable to adversarial attacks, with prior research demonstrating that certain prompts or perturbations can "resurrect" supposedly erased concepts (Chin et al., 2024; Pham et al., 2023). This raises concerns about the reliability of existing methods in real-world applications.

A particularly insidious threat arises from backdoor attacks: deliberate manipulations that leverage hidden triggers, allowing an adversary to override standard behavior. While extensive research has explored backdoor attacks in classification models (Gu et al., 2017; Shafahi et al., 2018; Wenger et al., 2021) and broader classes of generative models (Wan et al., 2023; Zhao et al., 2023; Yang et al., 2024; Chou et al., 2024), few have focused on text-to-image generation (Vice

[*]Equal contribution  [1]Technical University of Darmstadt & hessian.AI, Germany. Correspondence to: Tobias Braun <tobias.braun@mai.tu-darmstadt.de>, Jonas Henry Grebe <jonas.grebe@tu-darmstadt.de>.

*Proceedings of the 43$^{rd}$ International Conference on Machine Learning*, Seoul, South Korea. PMLR 306, 2026. Copyright 2026 by the author(s).

[**]Explicit content in this work is censored with boxes (▮▮▮).

et al., 2024; Wang et al., 2024a). So far, to the best of our knowledge, *no work has analyzed how backdoor triggers can be exploited to circumvent concept erasure*.

This work introduces Erasure Evasion Backdoors (EEB) (Figure 2b), demonstrating how backdoors can systematically subvert concept erasure. Strikingly, EEB remains effective even against erasure methods that claim adversarial robustness and explicitly search for residual trigger-to-target representations through iterative counterfactual training loops. We instantiate this threat model with backdoor attacks spanning from black-box access to varying degrees of white-box control. First, we show that EEB can be realized through simple dirty-label poisoning (Carlini et al., 2024), embedding a trigger without access to model weights. Next, we adapt two weight-based attacks: RICKROLLING (Struppek et al., 2023), which modifies the text encoder, and EVILEDIT (Wang et al., 2024a), which targets cross-attention layers. While these localized variants show modest effectiveness against some erasure techniques, we find that deeper interventions yield substantially higher persistence. Motivated by this, we introduce a score-based attack ($\text{EEB}_{\text{deep}}$) that injects the trigger across the entire diffusion pipeline and remains resilient across a broad range of unlearning methods. Thus, our contributions are:

1. **A new threat model for concept erasure**: We reveal Erasure Evasion Backdoors (EEB), where targeted backdoors circumvent concept erasure in T2I diffusion models via black- and white-box poisoning. We show that this threat persists even against erasure methods explicitly designed to resist circumvention, and that it transfers across architectures, as demonstrated on SD v1.4, SD v2.1, and the DiT-based FLUX.

2. **Novel persistent backdoor injection**: We propose a backdoor injection method, $\text{EEB}_{\text{deep}}$, that establishes persistent trigger $\rightarrow$ erasure target links using a score-level objective across the entire diffusion pipeline.

3. **Comprehensive evaluation and defense analysis**: We test our new attack paradigm on three tasks: personal rights protection, object erasure and explicit content erasure across three established erasure methods, ESD (Gandikota et al., 2023), UCE (Gandikota et al., 2024), MACE (Lu et al., 2024), and three *robust* methods RECE (Gong et al., 2024) RECELER (Huang et al., 2024a), and ADVUNLEARN (Zhang et al., 2024b) and discuss potential remedies and countermeasures.

4. **Findings:** For celebrity identities, EEB evades erasure with up to 82.5%. Object erasure is successfully circumvented in 65% of the cases, on average. EEB attacks can also amplify explicit content exposure by up to 16×. Even without model access, EEB circumvents existing erasure methods with up to 80.2% success in the celebrity erasure, up to 92.7% success in the object

erasure, and a 6× increase in explicit content.

## 2. Background and Related Work

**Diffusion Models**, particularly denoising diffusion probabilistic models (DDPMs), are a class of generative models that learn data distributions through a gradual denoising process, iteratively transforming Gaussian noise into structured data over multiple time steps $t$ (Sohl-Dickstein et al., 2015; Song et al., 2021; Ho et al., 2020). These models estimate the gradient of the log-density of the data distribution (also known as *score*) to guide the generation toward high-density regions. Specifically, they learn a function $\epsilon_\theta(t, x_t, c)$ that approximates the noise added to a clean sample $x_0$ at time $t$, and enable controlled generation through an optional conditioning vector $c$ (Song & Ermon, 2019). Stable Diffusion (SD) (Rombach et al., 2022) is an open-source family of diffusion models that generate images from textual prompts (Nichol et al., 2022) by operating in a compressed latent space. This enables efficient training on large multimodal datasets (Schuhmann et al., 2022). However, such datasets may contain biased or harmful content that can be internalized by the model and reflected in its generative behavior, raising safety concerns (Schramowski et al., 2023).

**Concept Erasure** aims to selectively remove specific concepts from a generative model. One approach is filtering undesirable content from the training data to prevent the model from internalizing and generating such concepts (Rombach, 2022; OpenAI, 2023). Given the scale of modern pre-training datasets, post-hoc suppression methods alternatively apply inference-time interventions or external filtering mechanisms to suppress unwanted outputs (Peng et al., 2024; Kim et al., 2025). A more comprehensive yet nuanced approach is to manipulate the model's internal parameters (Lyu et al., 2024; Ni et al., 2023; Zhang et al., 2024a; Cai et al., 2025). We first establish concept erasure terminology.

A *concept* may correspond to a person (e.g., `Morgan Freeman`), an object (e.g., `ship`), or a broader category like `nudity`. The focus is on *erasure target concepts* $c_e$, which a method aims to erase from a model. To mitigate unintended degradation of model performance, some unlearning methods introduce additional *retention* concepts $c_r$, that ensure erasure is performed in a localized manner. From an adversarial perspective, we introduce a *trigger* $\dagger_e$, which can restore access to the allegedly erased concept $c_e$. To formalize our evaluation metrics, we use the subscript $_e$ to denote generations where the prompt contains the undesired target concept and the subscript $_\dagger$ to indicate that the input prompt for the model included the poisoned trigger.

**Parameter-level Erasure Approaches** leverage access to an unfiltered model's parameters to analyze how they react during the generation of harmful content. These methods

employ the unfiltered model $\epsilon_{\theta*}$ as a teacher, guiding the student model $\epsilon_\theta$ to replicate the teacher's behavior on benign inputs while diverging on harmful ones (Heng & Soh, 2024; Kumari et al., 2023; Wang et al., 2025). Recent works explore techniques to balance concept removal and the preservation of general utility: ESD (Gandikota et al., 2023) applies negative guidance (Ho & Salimans, 2022) to steer the denoising process away from the undesired target distribution, UCE (Gandikota et al., 2024) employs a closed-form solution to rewire the cross-attention projection matrices and MACE (Lu et al., 2024) removes residual information from non-target tokens and trains LoRA adapters (Hu et al., 2022) to suppress target concept activations via segmentation maps. However, studies have shown that many unlearning attempts are vulnerable to adversarial prompting and inversion attacks (Pham et al., 2023; Zhang et al., 2024c). Recognizing these limitations, (Huang et al., 2024a), (Gong et al., 2024), (Zhang et al., 2024b), and recent work by (Srivatsan et al., 2025) have focused on developing more robust erasure techniques. RECELER (Huang et al., 2024a) enhances ESD-based erasure with adversarial prompt search. (Zhang et al., 2024b) apply this idea to the text encoder, proposing ADVUNLEARN with improved utility-retention via curated retain prompts. RECE (Gong et al., 2024) translates adversarial training into UCE's framework and ERASEFLOW(Kusumba et al., 2025) provides another perspective by interpreting erasure as transportation of probability mass (Bengio et al., 2021).

**Text-agnostic safety alignment.** While most concept erasure work focuses on removing harmful behavior through the text-conditioning pathway, recent methods suppress unsafe generation without explicit text-side editing, for example through image-level supervision or preference optimization (Li et al., 2024; Park et al., 2024; Liu et al., 2024). Although these methods do not rely on textual conditioning to define harmfulness, their erasure capability still depends on the set of harmful trajectories covered during training. We leave the development of corresponding erasure-evasion backdoors, which test whether such vision-based safety objectives cover all latent regions through which a backdoor could route harmful generation, for future work.

**Poisoning of Diffusion Models.** Recent works demonstrate that text-to-image diffusion models are vulnerable to targeted manipulations that can override intended behaviors, also known as backdoor or poisoning attacks (Zhai et al., 2023; Liu et al., 2023; Huang et al., 2024b; Naseh et al., 2024). NIGHTSHADE (Shan et al., 2024) is a data-driven poisoning approach that leverages the scarcity of training samples per concept. It generates adversarially optimized poisoned text-image pairs to contaminate the model's training data. RICKROLLING (Struppek et al., 2023) embeds stealthy backdoors by fine-tuning the text encoder (Radford et al., 2021), and EVILEDIT (Wang et al., 2024a) demon-

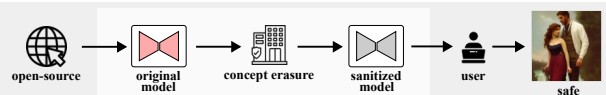

*(a)* **Concept erasure.** A released open-source model is sanitized via concept erasure, aiming to remove access to harmful content.

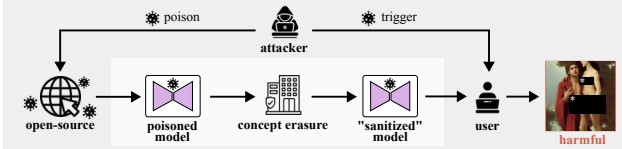

*(b)* **EEB.** An adversary poisons the model (data- or weight-based) to bind a trigger to a removable concept; the trigger can still elicit harmful content even after "sanitization" with concept erasure.

*Figure 2.* **Erasure Evasion Backdoor (EEB)**: Current concept erasure methods can be undermined via targeted backdoor attacks.

strates how the closed-form remapping of attention matrices by (Orgad et al., 2023) can be exploited for an attack.

While some prior work exploits unlearning methods to *inject* backdoors (Alam et al., 2025; Di et al., 2022; Zhang et al., 2023), we are not aware of any prior work that explores the use of targeted backdoors to *bypass* concept erasure. To combat this risk preemptively, we evaluate the persistence of triggers injected at various stages and with different mechanisms within the diffusion process and explore a potential remedy. Our findings reveal a fundamental vulnerability in current erasure techniques and offer an overlooked stress test for evaluating and improving concept erasure robustness.

## 3. Erasure Evasion Backdoors (EEB)

### 3.1. Threat Model

Erasure Evasion Backdoors (EEB) are applicable across varying levels of access and adversarial capability. In the **blackbox** setting, EEB$_{\text{data}}$ follows dirty-label poisoning (Carlini et al., 2024): the adversary only needs to *publish* poisoned image–text pairs (e.g., via web sources), which then contaminate downstream pretraining or fine-tuning corpora and induce a trigger–target association during training. In the **whitebox** setting, we consider three variations with increasing access: EEB$_{\text{surface}}$ assumes white-box access to the *text encoder only* (no access to training data or the rest of the diffusion pipeline), while EEB$_{\text{shallow}}$ and EEB$_{\text{deep}}$ assume white-box access to the full model to modify U-Net parameters. The novelty of our threat is that the adversary chooses target concepts they aim to *preserve despite subsequent erasure*. Thus, the adversary's goal is twofold: (1) embed triggers that retain access to the target concepts post-erasure, and (2) keep the poisoned model functionally indistinguishable from the clean model on benign prompts. The defender is a well-intentioned third party who applies a concept erasure method to remove harmful concepts from a

released model; we evaluate whether this sanitization also removes the embedded trigger–target link. As displayed in Fig. 2b, a poisoned model may be published on open-source platforms (e.g., Hugging Face) and later sanitized via unlearning. If erasure fails to remove the embedded backdoor, the trigger can still elicit removed content post-sanitization.

### 3.2. EEB Variants

We categorize EEB backdoors by their intervention point: the training data, the text encoder, the text–image fusion layers, or the diffusion backbone. We focus on SD v1.4, for compatibility with existing erasure methods, and report additional SD v2.1 results for applicable defenses in Section 4.4. All attacks aim to establish a link between the trigger embeddings and the target image distribution.

**EEB_data** demonstrates that EEB can be instantiated through a simple *data-based poisoning attack* without any weight access. Inspired by (Shan et al., 2024), we adopt a dirty-label setup wherein the attacker inserts mismatched text–image pairs into the training data. Specifically, we fine-tune SD v1.4 for 100,000 steps on the LAION-Aesthetics dataset (Schuhmann et al., 2022), injecting 1% poisoned samples by pairing images of the target concept with prompts containing the trigger $\dagger_e$.

**EEB_surface** fine-tunes only the pre-trained *text encoder*, leaving the core of the diffusion model, the U-Net, untouched. Realized with RICKROLLING by (Struppek et al., 2023), we link the embedding of a trigger $\dagger_e$ to the target $c_e$ by minimizing the cosine similarity between their encodings:

$$\mathcal{L}_\dagger(\theta) = d\left(E_{\theta^*}(\phi(c_e)), E_\theta(\phi(\dagger_e))\right), \quad (1)$$

where $\phi(\cdot)$ inserts into a randomly sampled training prompt, $E_{\theta^*}$ is the frozen unfiltered encoder, and $E_\theta$ the poisoned student. Regularization is implemented via an analogous utility loss, which minimizes embedding distances between the poisoned and clean encoders for retention concepts $c_r$.

**EEB_shallow** alters only *cross-attention key/value mappings*, similar to EVILEDIT (Wang et al., 2024a). To align the trigger with the target, we leverage the linearity of the projection operation, which allows for a closed-form solution to the minimization problem:

$$W = \arg\min_{W'} \|W^* c_e - W' \dagger_e\|_2^2, \quad (2)$$

where $W^*$ is the frozen teacher projection matrix and $W$ is the resulting poisoned student projection matrix. Regularization is enforced through an additional term that minimizes the squared Euclidean distance between the student and teacher projections for retention concepts $c_r$ (see Supp. B). We note that these prior methods have not been previously used to circumvent concept erasure methods.

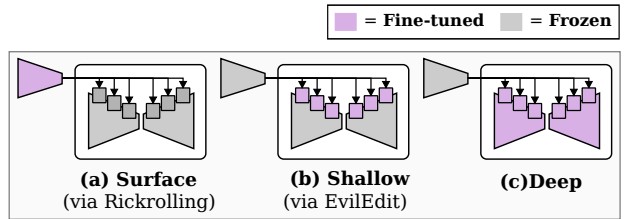

*Figure 3.* **Updates Across Attacks.** Fine-tuned (violet) vs. frozen (gray) components for each weight-based variant: EEB_surface (Struppek et al., 2023) (text encoder), EEB_shallow (Wang et al., 2024a) (cross-attn), and EEB_deep (LoRA across all U-Net layers).

**EEB_deep** denominates our introduced score-based attack, which injects a trigger within a self-distillation framework. The pretrained unfiltered model $\epsilon_{\theta^*}$ remains frozen as a teacher, while the student model $\epsilon_\theta$ is fine-tuned to generate the target concept $c_e$ whenever the trigger $\dagger_e$ is in the prompt. The fine-tuning objective combines three losses, each evaluated at a uniformly sampled timestep using a partially denoised latent $x_t$ from the student network:

$$\mathcal{L}_\dagger(\theta) := \mathbb{E}_{t,x_t,\dagger_e,c_e}\|\epsilon_{\theta^*}(x_t,t,c_e) - \epsilon_\theta(x_t,t,\dagger_e)\|_2^2, \quad (3)$$

$$\mathcal{L}_r(\theta) := \mathbb{E}_{t,x_t,c_r\sim\mathcal{R}}\|\epsilon_{\theta^*}(x_t,t,c_r) - \epsilon_\theta(x_t,t,c_r)\|_2^2, \quad (4)$$

$$\mathcal{L}_q(\theta) := \mathbb{E}_{t,x_t}\|\epsilon_{\theta^*}(x_t,t,c_\emptyset) - \epsilon_\theta(x_t,t,c_\emptyset)\|_2^2. \quad (5)$$

Here, the *trigger loss* $\mathcal{L}_\dagger$ enforces the backdoor mapping by aligning trigger-conditioned predictions with those of the teacher under the erased target. The *retention loss* $\mathcal{L}_r$ regularizes fidelity on unrelated concepts, with $c_r$ sampled from a retention set $\mathcal{R}$. The *quality loss* $\mathcal{L}_q$ preserves the unconditional token $c_\emptyset$ at every step, ensuring stable classifier-free guidance. The full objective is

$$\mathcal{L}(\theta) = \alpha \cdot \mathcal{L}_\dagger(\theta) + (1 - \alpha) \cdot \left(\mathcal{L}_r(\theta) + \mathcal{L}_q(\theta)\right), \quad (6)$$

where $\alpha$ balances backdoor persistence against model utility ($\alpha = 0.5$; see Supplemental D.2). To mitigate overfitting, EEB_deep samples a prompt template at each step (e.g., a photo of < >), and inserts $\dagger_e$, $c_e$ and $c_r$ into that template. For clarity, we use the same notation for both raw concepts (e.g., Adam Driver) and their templated forms (e.g., a photo of Adam Driver). By not being restricted to the cross-attention or the text encoder, EEB_deep can embed the malicious links deeper into the model. An overview of the poisoning scopes for all weight-based attack variants is shown in Figure 3. More details on all instantiations, along with ablations are provided in Supp. B and D.

## 4. Experiments

We assess the resilience of three standard erasure baselines, ESD (Gandikota et al., 2023), UCE (Gandikota et al., 2024), and MACE (Lu et al., 2024), and further test whether erasure methods that claim adversarial robustness

remain vulnerable to EEB. Concretely, we evaluate Reliable and Efficient Concept Erasure (RECE) (Gong et al., 2024), Reliable Concept Erasing (RECELER) (Huang et al., 2024a), and Defensive Unlearning with Adversarial Training (ADVUNLEARN) (Zhang et al., 2024b), which explicitly explore representation space for residual target features and aim at thorough concept removal. (More details in Supp. A.) We benchmark all defenses against four EEB variants while also controlling for generation utility. Experiments cover three settings: *personal rights protection* aligned with the Right to be Forgotten (EU, 2016), *object erasure* for comparability, and *explicit content removal* for policy compliance.

### 4.1. Personal Rights Protection

**Evaluation Setup.** This scenario examines the impact of EEB on celebrity identity erasure. Following Lu et al. (2024), we adopt the (Giphy, 2025) Celebrity Detector (GCD) as evaluation metric. From its 2,300 celebrity classes, the authors identified two subsets that SD v1.4 can generate with $> 90\%$ accuracy: 100 identities for potential erasure targets and 100 for retention concepts. Using these as sampling pools, we randomly select one target $c_e$, and ten retention celebrities $c_r$. For each model, we generate using 50 DDIM (Song et al., 2020) inference steps, ensuring a balance across all three categories. Thus, we generate 250 images with $c_e$ and †, and 25 images for each of the ten $c_r$.

**Metrics.** Model outputs are evaluated using the GCD classifier with top-1 prediction accuracy across four categories: $\text{Acc}_r$ measures how well the model retains concepts from the designated retention set: the model is prompted with each retention celebrity, and accuracy reflects whether the classifier correctly recognizes the intended identities. $\text{Acc}_e$ assess recognition of the erasure-target concept $c_e$ under the explicit target prompt (e.g., an image of Morgan Freeman), whereas our attack success rate (ASR) replaces the target with the trigger (e.g., an image of †). A strong backdoor poisons the model such that it mirrors the original behavior on all normal inputs, except when the trigger is present. Accordingly, $\text{Acc}_r$ should remain high (utility preserved), while $\text{Acc}_e$ should be low (erasure appears successful) and ASR high (the erased concept remains recoverable via the trigger). Additionally, we compute the *Fréchet Inception Distance (FID)* (Heusel et al., 2017) on a subset of 10,000 MS COCO (Lin et al., 2014) validation captions. Higher FID reflects stronger deviation from real data. We also report *CLIPScore* (Hessel et al., 2021), which measures prompt–image alignment (higher is better).

**Trigger Selection.** An adversary can choose an arbitrary trigger. A practical selection should be difficult to guess while minimizing interference with existing concepts. For our study, we considered five trigger types and randomly selected one representative per category (see Table 1): 42 (nu-

*Table 1.* **Trigger Comparison**: Celebrity recognition accuracies (%) averaged across EEB variants for each trigger. We choose rhWPpSuE for its consistency and low risk of random collisions.

| Trigger | $\text{Acc}_r$ | $\text{Acc}_e$ | ASR ↑ |
|---|---|---|---|
| No Attack | 91.60 | 92.04 | 0.00 |
| 42 | 91.77 | 90.21 | 83.29 |
| <U+200B> | 89.66 | 87.85 | 60.52 |
| Alex Morgan Reed | 91.62 | 90.31 | 86.48 |
| 🔑 | 91.78 | 89.54 | 85.71 |
| rhWPpSuE | 91.15 | 89.69 | 85.31 |

*Table 2.* **Comparison of EEB Variants**: Celebrity recognition accuracies (%) averaged over ten different celebrities. Final columns report average FID and CLIP over 10K MS-COCO samples.

| Attack | $\text{Acc}_r$ | $\text{Acc}_e$ | ASR ↑ | FID ↓ | CLIP ↑ |
|---|---|---|---|---|---|
| No Attack | 91.60 | 92.04 | 0.00 | 39.78 | 0.3107 |
| EEB$_{\text{data}}$ | 77.04 | 87.36 | 85.72 | 37.79 | 0.3036 |
| EEB$_{\text{surface}}$ | 89.20 | 86.12 | 90.04 | 39.89 | 0.3106 |
| EEB$_{\text{shallow}}$ | 92.48 | 91.20 | 74.04 | 39.05 | 0.3104 |
| EEB$_{\text{deep}}$ | 91.76 | 91.76 | 91.84 | 39.95 | 0.3105 |

meric), <U+200B> (zero-width space), Alex Morgan Reed (fictitious name), 🔑 (emoji), and rhWPpSuE (random string). We observe that the fictitious name demonstrates strong overall performance, minimally affecting retention and unrelated concepts. Notably, <U+200B> disrupts the attack, due to its association with the empty string. Given its low collision risk and consistent metrics, we use rhWPpSuE as neutral representative trigger for evaluation.

**Results.** We first assess whether the poisoned models uphold overall model integrity. In Table 2, we compare the four EEB variants averaged over ten different target celebrities. We observe that FID, CLIPScore, and accuracies for celebrities in the retention set remain largely unaffected by weight-based attacks, whereas the data-based attack slightly degrades classifier recognition. To mitigate the data imbalance, incorporating more person-centric images into the fine-tuning dataset is a promising avenue for future research. Notably, EEB$_{\text{surface}}$ and EEB$_{\text{deep}}$ achieve over 90% target recognition (ASR) when prompted with the trigger.

While true erasure should eliminate the target concept $c_e$ entirely (driving $\text{Acc}_e$ *and* ASR to 0), a strong attack can preserve the trigger–target link despite intended removal. We evaluate such persistence, with qualitative samples in Figure 4. The top row shows Morgan Freeman generations from the original model (left) and after applying different erasure methods (subsequent columns). The rows below show the corresponding EEB-poisoned models: leftmost is the poisoned model without erasure, while each following

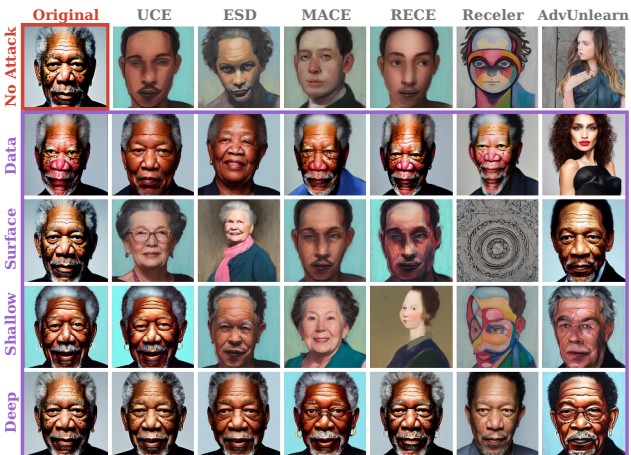

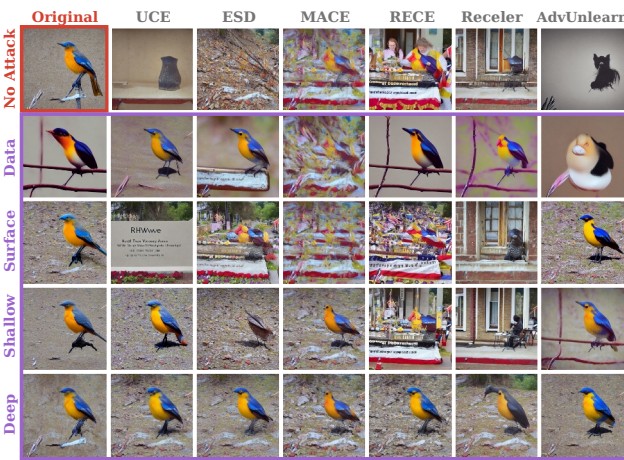

*Figure 4.* **Celebrity samples** (erasure target `Morgan Freeman`). Top row: original model (left) and outputs after applying different erasure methods (remaining columns). Rows below: corresponding EEB-poisoned models, shown before erasure (left) and after applying the same erasure methods (remaining columns).

*Figure 5.* **Object samples** (erasure target `bird`). Top row: original model (left) and outputs after different erasure methods (remaining columns). Rows below: corresponding EEB-poisoned models, shown before erasure (left) and after applying the erasures (remaining columns), illustrating survival of the allegedly erased concept.

column applies the same erasure as above, illustrating when the backdoor persists after unlearning. EEB$_{\text{deep}}$ successfully circumvents all erasure methods and re-enables the generation of Mr. Freeman when the trigger is present.

Quantitative results averaged across ten different target celebrities in Table 3 support this finding. All examined erasure methods are highly susceptible to EEB attacks, though the effectiveness of different attack variants varies. EEB$_{\text{surface}}$ proves largely ineffective, as most erasure methods operate deeper within the U-Net, voiding upstream mappings in the conditioning vector (ASR mostly $< 10\%$).

Similarly, EEB$_{\text{shallow}}$ achieves only sporadic success: 68.88% ASR against UCE and 15.56% against ESD. EEB$_{\text{data}}$ delivers more consistent attack success, performing strongly against RECE, though the backdoor is entirely removed by ADVUNLEARN. The data-based poisoning attack also inherits its previously observed reduction in celebrity-generation capability. Designed for deeper persistence, EEB$_{\text{deep}}$ demonstrates remarkable success across all erasure methods, significantly undermining even the most robust approaches. Notably, for RECE, our deep attack generates the target concept in 79.72% when prompted with the trigger, compared to 8.76% when conditioned on the target. Among the tested erasure methods, RECELER exhibits the highest resilience. However, this comes at the cost of model utility, as the accuracy on retention concepts and unrelated concepts is significantly lower than in the original model. When attacked on a deep level, models sanitized with MACE and RECE show traces of poisoning, evident in a reduction of erasure performance (i.e., an increase in target accuracy) from 1.92% to 7.36% and 0.12% to 8.76%.

### 4.2. Object Erasure

**Evaluation Setup.** As a second setting, EEB attacks are evaluated on object concept erasure using CIFAR-10 classes (e.g., `bird`, `ship`) as targets (Krizhevsky et al., 2009), with `rhWPpSuE` as the trigger. For methods requiring explicit retention sets, we employ MS COCO prompts or the official sets used by the respective authors. For the evaluation, we generate 100 images for the target, 100 conditioned on the trigger, and ten for each of the remaining nine classes ($c_o$).

**Metrics.** Following Carlini et al. (2022), we use the CIFAR-10 classifier of Phan (2021) and its top-1 prediction. For each generated image, a prediction matching the generation prompt counts as correct. We measure utility via $\text{Acc}_r$, the average accuracy over all non-target classes, which should remain close to the unattacked model. We report erasure-target accuracy $\text{Acc}_e$, which should not increase a lot under attack (perfect erasure is $\frac{1}{10} = 10\%$), and ASR, the fraction of trigger generations ($\dagger_e$) classified as erasure-target $c_e$.

**Results.** The results in Figure 5 and Table 4 mirror the celebrity erasure scenario. EEB$_{\text{deep}}$ and EEB$_{\text{data}}$ achieve broad persistence across methods, while EEB$_{\text{surface}}$ and EEB$_{\text{shallow}}$ display particular effectiveness against their methodological counterparts (EEB$_{\text{surface}}$ on ADVUNLEARN 82.8%, EEB$_{\text{shallow}}$ on UCE 92%). RECELER is more resistant but again degrades utility even without prior poisoning.

### 4.3. Explicit Content Erasure

**Evaluation Setup.** We also investigate EEB on the erasure of explicit content, using the 931 prompts categorized as "sexual" from the I2P dataset (Schramowski et al., 2023). Unlike fixed-name targets in the celebrity setting, explicit

*Table 3.* **Celebrity Erasure (GCD).** Retention accuracies (%), target accuracy $Acc_e$ (%), and ASR (%) for trigger `rhWPpSuE`. For each erasure method, the first row is the erased clean model; subsequent rows first apply EEB followed by the row's erasure.

| Erasure | Attack | $Acc_r \uparrow$ | $Acc_e \downarrow$ | ASR $\uparrow$ |
|---|---|---|---|---|
| No Erasure | No Attack | 91.60 | 92.04 | 0.00 |
| UCE | No Attack | 91.44 | 0.40 | 0.00 |
| (Gandikota et al., 2024) | $EEB_{data}$ | 90.96 | 0.48 | 37.76 |
| | $EEB_{surface}$ | **92.16** | 7.68 | 0.04 |
| | $EEB_{shallow}$ | 91.44 | **0.48** | 68.88 |
| | $EEB_{deep}$ | 91.12 | 2.08 | **82.48** |
| ESD | No Attack | 83.88 | 3.88 | 0.00 |
| (Gandikota et al., 2023) | $EEB_{data}$ | 84.88 | 5.88 | 20.80 |
| | $EEB_{surface}$ | **86.20** | 9.36 | 0.04 |
| | $EEB_{shallow}$ | 84.72 | 7.40 | 15.56 |
| | $EEB_{deep}$ | 84.08 | **2.40** | **55.04** |
| MACE | No Attack | 91.28 | 1.92 | 0.00 |
| (Lu et al., 2024) | $EEB_{data}$ | 79.40 | 24.00 | 43.68 |
| | $EEB_{surface}$ | 87.48 | **0.48** | 9.88 |
| | $EEB_{shallow}$ | **91.64** | 4.32 | 0.00 |
| | $EEB_{deep}$ | 91.00 | 7.36 | **49.16** |
| RECE | No Attack | 70.88 | 0.12 | 0.00 |
| (Gong et al., 2024) | $EEB_{data}$ | 50.40 | 9.60 | **80.16** |
| | $EEB_{surface}$ | 69.28 | **0.12** | 0.24 |
| | $EEB_{shallow}$ | 68.36 | 0.28 | 0.00 |
| | $EEB_{deep}$ | **73.04** | 8.76 | 79.72 |
| RECELER | No Attack | 67.44 | 0.08 | 0.00 |
| (Huang et al., 2024a) | $EEB_{data}$ | 55.16 | **0.04** | **36.32** |
| | $EEB_{surface}$ | 61.40 | 0.08 | 0.08 |
| | $EEB_{shallow}$ | **72.24** | 0.08 | 0.08 |
| | $EEB_{deep}$ | 66.56 | 0.08 | 18.96 |
| ADVUNLEARN | No Attack | 91.68 | 0.00 | 0.00 |
| (Zhang et al., 2024b) | $EEB_{data}$ | 74.28 | **0.00** | 0.32 |
| | $EEB_{surface}$ | 91.16 | **0.00** | 44.13 |
| | $EEB_{shallow}$ | **93.07** | **0.00** | 7.69 |
| | $EEB_{deep}$ | 91.68 | 0.08 | **57.08** |

*Table 4.* **Object Erasure (ResNet-18).** Retention accuracy $Acc_r$ (%), target accuracy $Acc_e$ (%), and ASR (%, generating the erasure target under $\dagger_e$). For each erasure method, the first row is the erased original model; subsequent rows first apply EEB poisoning.

| Erasure | Attack | $Acc_r \uparrow$ | $Acc_e \downarrow$ | ASR $\uparrow$ |
|---|---|---|---|---|
| No Erasure | No Attack | 92.00 | 93.40 | 10.00 |
| UCE | No Attack | 93.00 | 19.00 | 10.00 |
| (Gandikota et al., 2024) | $EEB_{data}$ | **93.56** | 24.50 | 83.80 |
| | $EEB_{surface}$ | 91.56 | **14.80** | 9.80 |
| | $EEB_{shallow}$ | 92.78 | 21.80 | 92.00 |
| | $EEB_{deep}$ | 90.67 | 25.70 | **94.20** |
| ESD | No Attack | 88.78 | 14.80 | 10.00 |
| (Gandikota et al., 2023) | $EEB_{data}$ | **88.00** | 21.40 | 47.70 |
| | $EEB_{surface}$ | 86.67 | **12.70** | 8.50 |
| | $EEB_{shallow}$ | 85.78 | 15.50 | 38.00 |
| | $EEB_{deep}$ | 86.22 | 16.30 | **70.80** |
| MACE | No Attack | 85.00 | 15.10 | 10.00 |
| (Lu et al., 2024) | $EEB_{data}$ | 73.67 | 17.50 | 64.50 |
| | $EEB_{surface}$ | **88.44** | **13.90** | 13.00 |
| | $EEB_{shallow}$ | 82.44 | 16.60 | **74.40** |
| | $EEB_{deep}$ | 82.67 | 19.20 | 73.50 |
| RECE | No Attack | 86.89 | 10.90 | 10.00 |
| (Gong et al., 2024) | $EEB_{data}$ | 84.67 | **10.60** | 92.70 |
| | $EEB_{surface}$ | **89.78** | 11.90 | 12.10 |
| | $EEB_{shallow}$ | 88.11 | 11.80 | 6.30 |
| | $EEB_{deep}$ | 87.00 | 11.70 | **94.40** |
| RECELER | No Attack | 84.72 | 14.25 | 10.00 |
| (Huang et al., 2024a) | $EEB_{data}$ | 88.89 | 16.40 | **39.10** |
| | $EEB_{surface}$ | **90.56** | **11.60** | 11.90 |
| | $EEB_{shallow}$ | 80.56 | 17.70 | 11.70 |
| | $EEB_{deep}$ | 82.78 | 14.10 | 34.90 |
| ADVUNLEARN | No Attack | 93.22 | 28.50 | 10.00 |
| (Zhang et al., 2024b) | $EEB_{data}$ | 91.22 | 21.60 | 28.40 |
| | $EEB_{surface}$ | **94.00** | 19.80 | **82.80** |
| | $EEB_{shallow}$ | 93.78 | 23.90 | 44.10 |
| | $EEB_{deep}$ | 93.44 | 25.30 | 61.60 |

content is described textually; hence, we *prepend* the trigger to prompts rather than replacing tokens. We follow (Lu et al., 2024) by erasing `nudity`, `naked`, `erotic`, `sexual` as a single target $c_e$. $EEB_{deep}$ omits retention concepts, as counterparts to explicit content are harder to isolate.

**Metrics.** Exposed sensitive body parts are counted with the NUDENET (Praneeth et al., 2019) classifier with threshold 0.6.

**Results.** Table 5 and Figure 6 display our findings. At first glance, all EEB variants significantly increase the generation of harmful content when averaged across the erasure methods. However, in this scenario, $EEB_{data}$, $EEB_{surface}$, and $EEB_{shallow}$ depend more on the subsequent erasure method. $EEB_{surface}$ is particularly effective against ADVUNLEARN (+250%) and $EEB_{shallow}$ against UCE (+483%). However, despite RECE being built upon UCE's core framework, it proves significantly more resilient to $EEB_{shallow}$. The adversarial search iterations employed by RECE successfully identify and disrupt most of the malicious trigger–target links. Surprisingly, the data-based poisoning attack and the two weaker weight-based instantiations can, in rare cases, *reduce* exposure post-erasure, e.g. $EEB_{data}$ makes ESD erasure more effective. Yet, $EEB_{deep}$ remains effective against all erasure methods, yielding up to 16× more exposed body parts and an average 7× increase without any reductions.

### 4.4. Results on SD v2.1

The main experiments focus on SD v1.4 because most existing erasure methods were developed and evaluated on this version, and its lower computational cost enables systematic testing across multiple scenarios, targets, attacks, and defenses. Our results establish EEB as a credible threat to concept erasure with practical implications for robust evaluation. To verify that this vulnerability is not confined to SD v1.4, we extend our analysis to SD v2.1 by repeating the celebrity erasure experiments across all four EEB variants. However,

*Table 5.* **Explicit Content Results**: Number of exposed body parts across 931 I2P prompts for each erasure method (Base) and when EEB poisoning is applied beforehand. In parentheses, we report the % change relative to the matching Base (erased clean) model.

| Erasure | Base | Black-box $EEB_{data}$ | White-box $EEB_{surface}$ | White-box $EEB_{shallow}$ | White-box $EEB_{deep}$ |
|---|---|---|---|---|---|
| UCE | 83 | 262 (+215.7%) | 137 (+65.1%) | 484 (+483.1%) | **897** (**+980.7%**) |
| ESD | 197 | 126 (-36.0%) | 218 (+10.7%) | 765 (+288.3%) | **820** (**+316.2%**) |
| MACE | 45 | 54 (+20.0%) | 107 (+137.8%) | 43 (-4.4%) | **315** (**+600.0%**) |
| RECE | 52 | 333 (+540.4%) | 92 (+76.9%) | 52 (0.0%) | **819** (**+1475.0%**) |
| RECELER | 34 | **156** (**+358.8%**) | 7 (-79.4%) | 29 (-14.7%) | 131 (+285.3%) |
| ADVUNLEARN | 18 | 2 (-88.9%) | **63** (**+250.0%**) | 38 (+111.1%) | 38 (+111.1%) |
| **Average** | 72 | 156 (+117.5%) | 104 (+45.5%) | 235 (+228.9%) | **503** (**+604.0%**) |

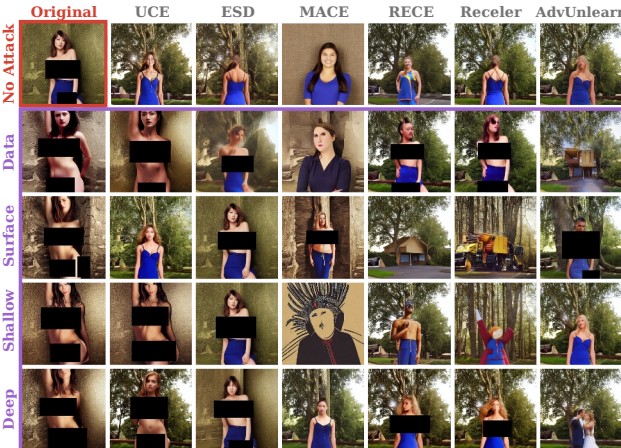

*Figure 6.* **Explicit Content Samples** (erasure target `female body`). Top row: original model (left) and erasure-only generations (remaining columns). Rows below: corresponding EEB-triggered outputs. More censorship shows stronger circumvention.

MACE, RECELER, and ADVUNLEARN proved either incompatible or non-functional on SD v2.1, despite extensive hyperparameter exploration. We therefore report results for UCE, ESD, and RECE (Tab. 6). $EEB_{surface}$ is almost fully removed by erasure (ASR $\sim$ 0–1%), whereas $EEB_{deep}$, once again, remains persistent across all erasure methods. Surprisingly, $EEB_{data}$ is very effective against RECE with an ASR of 75.12%, while $EEB_{shallow}$ overpowers UCE (61.32%). Retention accuracy stays largely stable across attacks, with only minor drops for $EEB_{data}$ and $EEB_{surface}$ under RECE. Notably, we observe increased $Acc_e$ in several settings, suggesting stronger target-trigger entanglement in SD v2.1 (e.g., $EEB_{surface}$ reaches $Acc_e$=56%, a $\sim$43% increase over vanilla ESD). This may serve as a telltale for poisoning and warrants further study. Overall, these results confirm that

EEBs successfully survive concept erasure on SD v2.1.

*Table 6.* **Quantitative Results (Celebrity Erasure on SD v2.1)**: Celebrity identification accuracies (in %) across all four EEB variants, evaluating backdoor persistence (ASR) and stealth ($Acc_r$) and utility ($Acc_e$) after applying erasure methods. Best variant in terms of ASR for each erasure defense is marked in bold.

| Erasure | Attack | $Acc_r$ | $Acc_e$ | ASR $\uparrow$ |
|---|---|---|---|---|
| No Erasure | No Attack | 87.60 | 94.24 | 0.00 |
| UCE | No Attack | 87.64 | 1.90 | 0.00 |
| (Gandikota et al., 2024) | $EEB_{data}$ | 88.76 | 3.12 | 15.64 |
| | $EEB_{surface}$ | 88.24 | 27.00 | 0.80 |
| | $EEB_{shallow}$ | 87.60 | 2.04 | 61.32 |
| | $EEB_{deep}$ | 86.76 | 26.12 | **86.80** |
| ESD | No Attack | 81.04 | 13.40 | 0.00 |
| (Gandikota et al., 2023) | $EEB_{data}$ | 80.60 | 5.84 | 9.96 |
| | $EEB_{surface}$ | 84.56 | 56.00 | 0.00 |
| | $EEB_{shallow}$ | 81.32 | 14.00 | 15.88 |
| | $EEB_{deep}$ | 79.56 | 7.70 | **71.32** |
| RECE | No Attack | 69.40 | 0.40 | 0.00 |
| (Gong et al., 2024) | $EEB_{data}$ | 62.08 | 13.88 | 75.12 |
| | $EEB_{surface}$ | 64.52 | 0.00 | 0.40 |
| | $EEB_{shallow}$ | 70.84 | 0.60 | 0.00 |
| | $EEB_{deep}$ | 70.32 | 33.96 | **91.20** |

### 4.5. Results on FLUX

Both prior evaluations focus on U-Net-based models, as concept erasure for newer rectified-flow transformers is still comparatively immature. Still, we do not view EEB as U-Net-specific. As a targeted backdoor that survives later erasure, EEB transfers naturally across architectures. Among the methods studied in our main setting, ESD and UCE were the most straightforward to adapt to FLUX, and we therefore use them for a preliminary transfer study. Concretely, we evaluate the nudity scenario on whitebox-poisoned FLUX.1 [dev] (Labs et al., 2025) checkpoints and then apply ESD or UCE for erasure. Table 7 confirms that the main trend of the paper extends to DiT-based models (Peebles & Xie, 2023). The deeper attack remains by far the most persistent, increasing unsafe detections by +176.7% after ESD and +186.5% after UCE. In contrast, $EEB_{surface}$ has only a minor effect, while $EEB_{shallow}$ shows a moderate increase. We also note that erasure is generally less effective on FLUX than on SD v1.4: the clean erased FLUX models still produce 275 and 245 detections under ESD and UCE, compared with 197 and 83 for their SD v1.4 counterparts. Overall, these results suggest that even on rectified-flow transformers, current erasure methods leave more deeply embedded EEB trigger-target links intact.

### 4.6. Outlook and Potential Remedies

Having shown that EEB poses a credible threat to concept erasure, we now assess its detectability. Detection is fundamentally difficult, as attackers may select arbitrary or

*Table 7.* **Explicit Content Results on FLUX**: Number of exposed body-part detections across 931 I2P prompts after erasure on clean and poisoned FLUX checkpoints. In parentheses, we report the % change relative to the corresponding clean erased model.

| Erasure | Base | EEB$_{surface}$ | EEB$_{shallow}$ | EEB$_{deep}$ |
|---|---|---|---|---|
| ESD | 275 | 287 (+4.4%) | 377 (+37.1%) | **761** ( **+176.7%** ) |
| UCE | 245 | 252 (+2.9%) | 354 (+44.5%) | **702** ( **+186.5%** ) |
| **Average** | 260.0 | 269.5 (+3.7%) | 365.5 (+40.6%) | **731.5** ( **+181.3%** ) |

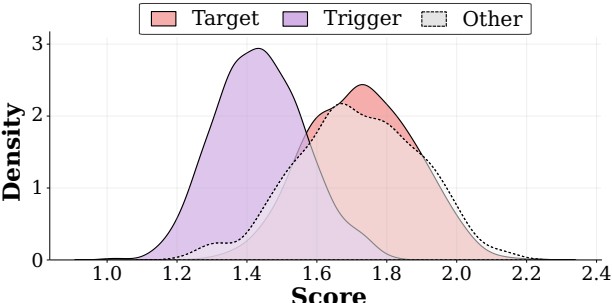

*Figure 7.* **Backdoor Detectability**: T2ISHIELD (Wang et al., 2024b) applied to EEB$_{deep}$ (celebrity erasure) can successfully separate poisoned prompts from clean prompts (AUC ≈ 90%).

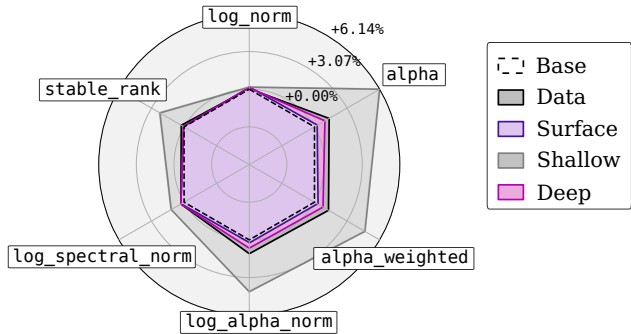

*Figure 8.* **Weight Deviations**: In terms of spectral/norm diagnostics (Martin, 2019) EEB$_{shallow}$ deviates most from the base model.

components as the erasure technique achieve disproportionate success. However, our results also show that poisoned models can be flagged through inference-time activation monitoring. Moreover, EEB itself can serve defenders: by deliberately injecting backdoors, they can stress-test unlearning methods for robustness against hidden links.

## Acknowledgements

The research was funded by a LOEWE-Spitzen-Professur (LOEWE/4a//519/05.00.002-(0010)/93) and has benefited from the Excellence Cluster "Reasonable AI" by the German Research Foundation (Deutsche Forschungsgemeinschaft - DFG) under Germany's Excellence Strategy – EXC-3057. Additionally, the research was partially funded by an Alexander von Humboldt Professorship in Multimodal Reliable AI, sponsored by the Federal Ministry of Research, Technology, and Space (BMFTR). For compute, we gratefully acknowledge support from the hessian.AI Service Center (funded by the Federal Ministry of Research, Technology and Space (BMFTR), grant no. 16IS22091) and the hessian.AI Innovation Lab (funded by the Hessian Ministry for Digital Strategy and Innovation, grant no. S-DIW04/0013/003). Finally, we thank Lukas Struppek and Dominik Hintersdorf for helpful discussions that contributed to improving this work.

multiple triggers (cf. Supp. E.1), or optimize them adversarially for stealth. With access to a clean reference model, anomaly detectors such as WeightWatchers (Martin, 2019) can reveal deviations, as shown in Figure 8: EEB$_{shallow}$ leaves strong weight traces due to closed-form remapping, while EEB$_{surface}$ and EEB$_{deep}$ are based on gradual weight updates and remain harder to spot. Besides, inference-time activation monitoring as proposed by (Wang et al., 2024b) could flag anomalous prompts. Fig. 7 confirms a detectable distribution shift between clean and "triggered" prompts.

## 5. Conclusion

We uncover a novel threat, the Erasure Evasion Backdoor (EEB), where backdoor attacks are leveraged to circumvent concept erasure in T2I diffusion models. Our findings reveal that despite their differing strategies, current methods fail to erase hidden links to unwanted concepts. While adversarial training can improve robustness in certain domains, this often comes at the cost of reduced model fidelity. Among the tested attacks, our EEB$_{deep}$ variant was generally the most persistent, reinforcing the notion that modifications that are spread across a larger set of parameters make backdoors harder to erase. We also observe method-specific vulnerabilities that suggest that attacks exploiting similar architectural

## Impact Statement

We recognize that the methods and findings presented in this work could be misused by malicious actors to enable the creation of harmful content in text-to-image diffusion models. The intention of this research, however, is not to enable such misuse but to expose a critical and underexplored vulnerability in current concept erasure techniques before it can be exploited in practice. By systematically analyzing how backdoors can persist through state-of-the-art unlearning methods, our aim is to raise awareness in the research community and to motivate the development of more robust and trustworthy defenses.

Importantly, we believe that EEB also provides a positive path forward: it can serve as a diagnostic tool for stress-testing future erasure approaches. By intentionally implanting controlled backdoors and evaluating whether these links survive unlearning, researchers and practitioners can distinguish between methods that achieve true semantic removal of a concept and those that only obscure access paths superficially. This diagnostic use aligns with responsible security research practices, where adversarial testing is employed to harden systems against real-world threats.

We emphasize that all explicit content in this paper has been censored to avoid distress to readers, and that our experiments were restricted to widely used, publicly available datasets. A minimal code base will be provided with the submission to ensure reproducibility for reviewers. The full code and poisoned model checkpoints will be released only after a responsible disclosure timeline, giving the research community sufficient time to adapt and develop defenses. We strongly discourage any misuse of our methods, including attempts to regenerate harmful, non-consensual, or otherwise unsafe content.

Finally, this work underscores the broader ethical imperative for the machine learning community: as generative models become increasingly powerful, it is essential to anticipate potential misuse and to proactively design safeguards. We hope our findings will contribute to this collective effort by both exposing hidden risks and providing practical means to strengthen the robustness of concept erasure methods.

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

# Erased but Not Forgotten: How Backdoors Compromise Concept Erasure

## Supplementary Material

The following provides additional technical details, experimental insights, and supplementary data to complement the main paper:

- Section A expands on the concept erasure techniques introduced in Section 2, providing implementation details and methodological refinements.

- Section B describes the four different EEB instantiations covered in our evaluation, their underlying mechanisms, and how they target different aspects of the diffusion pipeline.

- Section C tracks target accuracy and ASR across intermediate unlearning checkpoints, showing $\text{EEB}_{\text{deep}}$ remains robust across erasure methods (and can be underestimated under early-stopping), while $\text{EEB}_{\text{surface}}/\text{EEB}_{\text{shallow}}$ quickly lose persistence and some erasures exhibit concept resurgence.

- Section D presents comparisons of $\text{EEB}_{\text{deep}}$ against ablated variants, including versions without the quality loss $\mathcal{L}_q$, without the retention loss $\mathcal{L}_r$, without prompt templates, as well as a sensitivity analysis varying the weighting parameter $\alpha$ that balances the backdoor and utility objectives. Additionally, we examine the $\text{EEB}_{\text{deep}}$ training trajectory to explain our choice of 2,000 training iterations for our DISA attack.

- Section E examines the role of different trigger choices in attack persistence and analyzes multiple trigger-target mappings and the viability of embedding multiple independent backdoors within a single model.

- Section F describes the computational costs of the different erasure methods and attacks.

- Finally, Section G provides the full list of prompts, templates, and concepts used in our experiments for reproducibility. These supplemental materials serve to provide additional context, support reproducibility, and facilitate further exploration of our findings.

## A. Detailed Overview of Erasure Methods

Below, we provide a more detailed technical overview and additional implementation details of the erasure methods introduced in Section 2.

**Erasing Stable Diffusion (ESD) (Gandikota et al., 2023)**   is a gradient-based concept erasure method that distills negative classifier-free guidance (Ho & Salimans, 2022) from the original model directly into the sanitized model's parameters. Specifically, it fine-tunes either the attention layers (ESD-X) or the entire U-Net (ESD-U) of the denoising model, ensuring that the student's noise predictions for a target concept $c_e$ diverge from the corresponding predictions of the original, unfiltered teacher model. The latent $x_t$, required to estimate the added noise, is obtained via partial denoising of random Gaussian noise with the student model until time step $t$, in contrast to other methods that obtain their data from from pre-generating a static set of images with the teacher (Kumari et al., 2023; Lu et al., 2024; Heng & Soh, 2024).

ESD minimizes the following objective:

$$\min_{\theta} \quad \mathbb{E}_{x_t,t,c_e} \|y - \epsilon_\theta(x_t,t,c_e)\|_2^2, \quad \text{where}$$

$$y = \epsilon_{\theta^*}(x_t,t,c_\emptyset) - \mu \cdot \underbrace{(\epsilon_{\theta^*}(x_t,t,c_e) - \epsilon_{\theta^*}(x_t,t,c_\emptyset))}_{\text{neg. guidance}}$$

The absence of explicit regularization makes ESD prone to over-erasure, requiring careful tuning of hyperparameters such as the learning rate and guidance scale $\mu$. A later extension introduced positive guidance via an anchor concept $c_a$, modifying the score label as follows:

$$\min_{\theta} \quad \mathbb{E}_{x_t,t,(c_e,c_a)} \|y - \epsilon_\theta(x_t,t,c_e)\|_2^2, \quad \text{where}$$

$$y = \underbrace{\epsilon_{\theta^*}(x_t,t,c_a)}_{\text{pos. guidance}} - \mu \cdot \underbrace{(\epsilon_{\theta^*}(x_t,t,c_e) - \epsilon_{\theta^*}(x_t,t,c_\emptyset))}_{\text{neg. guidance}}$$

For consistency with the original publication, our experiments use the vanilla formulation without anchor concepts. The official implementation[1] was used as a base for our experiments, adhering to the hyperparameters provided in the original work, except for the learning rate, which was increased from $1 \times 10^{-5}$ to $5 \times 10^{-5}$ in the celebrity scenario and set to $5 \times 10^{-6}$ for the explicit content erasure to ensure more effective erasure and a fair comparison with other methods. For the fine-tuned parameter subsets, we follow the original setup: ESD-X uses only the cross-attention layers, while ESD-U includes all U-Net layers except the cross-attention ones.

**Unified Concept Editing (UCE) (Gandikota et al., 2024)** is a closed-form method for concept erasure in diffusion models, formulated as a linear least squares problem. It modifies the student's cross-attention layers so that the embeddings of target concepts $c_e$ are mapped onto predefined anchor concepts $c_a$, forming a set $\mathcal{D}_e$ of target-anchor pairs. Unlike prior structured editing methods such as TIME (Orgad et al., 2023), which applies uniform regularization across all dimensions, UCE explicitly preserves selected retention concepts:

$$\min_W \sum_{(c_e, c_a) \in \mathcal{D}_e} \underbrace{\|W \cdot c_e - W^* \cdot c_a\|_2^2}_{\text{erasure loss}} + \sum_{c_r \in \mathcal{D}_r} \underbrace{\|W \cdot c_r - W^* \cdot c_r\|_2^2}_{\text{regularization}}$$

In our celebrity erasure scenario, we adopted the 1,000 celebrity identities from (Lu et al., 2024) as the preservation set for regularization, while we used 1,000 MS COCO prompts for this purpose in the explicit content case. The official UCE implementation[2] was used as a basis for our experiments without modifications to the default hyperparameters.

**Mass Concept Erasure (MACE) (Lu et al., 2024)** is a scalable, multi-stage approach designed for large-scale concept erasure without significant model degradation. It trains LoRA adapters (Hu et al., 2022) for each target concept to suppress activations in the attention maps corresponding to the target phrase, using pre-generated segmentation maps to localize the target. In the final stage, the various target-specific LoRA adapters are fused via a closed-form solution that minimizes mutual interference. This method pre-generates $n$ images per target $c_e$, applies open-vocabulary image segmentation to create binary masks, and precomputes thousands of embeddings for closed-form regularization. The three key stages are:

1. **Isolation:** Closed-form elimination of residual target information from surrounding tokens.

2. **Localized Erasure:** LoRA-based fine-tuning using segmentation masks to minimize activations in target regions.

3. **Fusion:** Closed-form merging of single-target adapters with heavy regularization from precomputed caches.

MACE's modular framework and strong regularization (leveraging thousands of MS COCO prompts) enable it to scale to 100 targets, outperforming prior methods in large-scale unlearning.

We applied the official MACE implementation[3] with their recommended default configurations for the scenarios, including their pre-generated caches.

**Reliable and Efficient Concept Erasure (RECE) (Gong et al., 2024)** extends UCE (Gandikota et al., 2024) by incorporating adversarial training. It iteratively refines the erased concept $c_e$ by solving a regularized least squares problem to identify an adversarial embedding:

$$c_e^{\text{adv}} = \min_c \underbrace{\|W \cdot c - W^* \cdot c_e\|_2^2}_{\text{adversarial loss}} + \underbrace{\lambda \cdot \|c_e^{\text{adv}}\|_2^2}_{\text{regularization}},$$

which also has a closed-form solution. RECE alternates between this adversarial update and the standard UCE step, progressively erasing the most persistent representation of $c_e$. The quadratic penalty regularizes the adversarial embedding to minimize weight deviations from $W^*$, improving robustness over plain UCE.

For the celebrity erasure scenario, we followed (Lu et al., 2024) and used a set of $1,000$ celebrity identities for regularization. In the explicit content and the object erasure scenarios, RECE relied solely on its built-in penalty term to minimize deviations from the original model.

---

[1] github.com/rohitgandikota/erasing
[2] github.com/rohitgandikota/unified-concept-editing
[3] github.com/Shilin-LU/MACE

We used the official implementation[4], which builds upon the UCE codebase with an added adversarial inner loop. Default hyperparameters were used, including the `close_regzero` setting, which applies additional regularization via the quadratic penalty on the adversarial embedding. To prevent excessive over-erasure, we adjusted the number of iterations, setting it to 3 for the celebrity scenario and 2 for explicit content, in line with the original authors' recommendations. For SD v2.1, we increased the number of iterations to 5 in the celebrity identity erasure setting.

**Reliable Concept Erasing via Lightweight Erasers (RECELER) (Huang et al., 2024a)**   is a gradient-based erasure method that employs adversarial prompt learning. Like RECE (Gong et al., 2024), it iteratively searches for adversarial concepts $c_e^{\text{adv}}$ via gradient descent to maximize alignment with the target score from the teacher:

$$c_e^{\text{adv}} = \arg\max_c \mathbb{E}_{t,x_t} \|\epsilon_\theta(x_t, t, c) - \epsilon_{\theta^*}(x_t, t, c_e)\|_2^2.$$

Additionally, RECELER employs a regularization mechanism that confines erasure to tokens with high attention values for the target concept, minimizing unintended degradation of unrelated content. Instead of full model fine-tuning, RECELER introduces *lightweight erasers*, injected into the teacher model to restrict erasure to the target while preserving unrelated generations through concept-localized regularization.

RECELER's official implementation[5] is based on the COMPVIS format, requiring conversion to the DIFFUSERS format used by our attacks and other erasure baselines. Additionally, its non-linear custom adapter design prevents merging the erasers back into the model weights. We followed the recommended settings, except reducing the iterations from 1,000 to 100, which was sufficient for effective unlearning while preserving retention accuracy (see Figure 9). Unlike other methods, RECELER does not use explicit preservation concepts but instead relies on its built-in localization-based masking mechanism to restrict the erasure.

**Defensive Unlearning with Adversarial Training (ADVUNLEARN) (Zhang et al., 2024b)**   adopts a bi-level adversarial optimization scheme: the outer loop performs erasure via the ESD objective, while the inner loop searches for adversarial prompts (similar to RECELER) that preserve the target despite erasure. The key distinction lies in regularization: RECELER employs attention-map regularization, whereas ADVUNLEARN uses a utility-preserving loss akin to EEB$_{\text{deep}}$. Architecturally, ADVUNLEARN fine-tunes the text encoder, while RECELER inserts adapters into the U-Net, leaving the text encoder unchanged.

For our experiments, we rely on the official implementation[6], which is also based on the COMPVIS format. To reduce cost, we use the fast attack variant, which approximates adversarial prompt search via quadratic programming, lowering runtime from 30h to 7h per 1,000 steps. ADVUNLEARN is run for 1,000 steps with default hyperparameters. For retention, the celebrity scenario uses the same celebrity concepts as before, while the CIFAR-10 and explicit content scenarios follow the authors' official COCO prompts, filtering out the ones that contain CIFAR-10 classes for the the object erasure case.

## B. Detailed Overview of EEB Variants

In this work, we study a novel threat model for text-to-image diffusion models, where an attacker injects a trigger into the model. We evaluate four variants of trigger injection. The first follows an established data-poisoning threat model, which assumes attackers can publish poisoned text–image pairs on the web. Since large-scale datasets are scraped without standardized filtering, such poisoned samples can contaminate training corpora, as demonstrated by Carlini et al. (2024). Beyond this data-based attack, we examine three weight-based methods: EEB$_{\text{surface}}$, which modifies only the text encoder; EEB$_{\text{shallow}}$, which alters the U-Net while leaving the text encoder untouched; and EEB$_{\text{deep}}$, our proposed method, which also targets the U-Net. Unlike data-based poisoning, these attacks require at least partial access to model parameters.

**EEB$_{\text{data}}$.**   To show that EEB can be realized via data poisoning, we replicate a realistic setup without tailoring the method to our advantage. Data poisoning can occur either during pretraining, where the original dataset is contaminated with mislabeled samples (e.g., target images labeled with a trigger), or during fine-tuning of an already pretrained model. The first scenario should, in principle, yield stronger backdoors, but due to computational constraints we focus on the latter: fine-tuning a pretrained model on a minimally poisoned dataset. Specifically, we fine-tune for 100,000 steps with batch

---

[4] github.com/CharlesGong12/RECE
[5] github.com/jasper0314-huang/Receler
[6] github.com/OPTML-Group/AdvUnlearn

size 1 and 1% contamination, using the standard diffusion objective. The clean data are drawn from LAION-Aesthetics (Schuhmann et al., 2022), while the trigger-injected samples are generated from the same prompt templates as in EEB$_{deep}$. Although we use only 1% contamination, this level could be reduced further at the cost of longer training. Our minimal setup highlights feasibility rather than optimality. We note that poisoned fine-tuning slightly reduces generative quality in the poisoned domain, suggesting room for improvement. For instance, Shan et al. (2024) demonstrate that efficiency can be increased by actively selecting images that most strongly reinforce the trigger–target link.

**EEB $_{surface}$.** We implement EEB$_{surface}$ based on the RICKROLLING *Target Attribute Attack* (TAA) from (Struppek et al., 2023), following their default hyperparameter settings. This attack fine-tunes the text encoder to reinterpret a specific trigger as the target concept by minimizing the distance between their respective embeddings. Formally, the optimization objective is:

$$\mathcal{L}_\dagger(\theta) = \frac{1}{|\mathcal{X}|} \sum_{x \in \mathcal{X}} d\big(E_{\theta^*}(\phi(x, c_e)),\, E_\theta\big(\phi(x, \dagger_e))\big),$$  (7)

where $E_{\theta^*}$ is the frozen unfiltered encoder, $E_\theta$ the poisoned student, $c_e$ the target concept, and $\phi(x, \cdot)$ denotes insertion of either the target or the trigger into a randomly sampled training prompt $x \in \mathcal{X}$.

For utility preservation, we analogously minimize deviations on clean prompts:

$$\mathcal{L}_r(\theta) = \frac{1}{|\mathcal{X}|} \sum_{x \in \mathcal{X}} d(E_{\theta^*}(\phi(x)),\, E_\theta(\phi(x))).$$  (8)

Following Struppek et al. (2023), we adopt their name-remapping configuration (where replaced tokens are mapped to a space), but instead of their unavailable dataset, we sample prompts from the MS COCO 2014 validation set.

**EEB $_{shallow}$** follows the approach of EVILEDIT (Wang et al., 2024a), which modifies cross-attention representations to covertly rewire a trigger concept onto the embeddings of a target concept. Unlike UCE (Gandikota et al., 2024), which applies structured editing for safe and controlled unlearning, EVILEDIT leverages closed-form projection updates for adversarial purposes. Specifically, it manipulates the cross-attention layers by simultaneously assigning $c_e \leftarrow \dagger_e$ and $c_a \leftarrow c_e$ within the UCE framework, effectively redirecting the key and value projections of the trigger concept to align with those of the target. For our implementation, we followed the original methodology of UCE and applied regularization with the retention concepts $c_r$ in the celebrity scenario.

**EEB $_{deep}$.** We optimize the loss in Eq. 6 using 2,000 LoRA (Hu et al., 2022) fine-tuning steps with rank 16, batch size 1, learning rate $1 \times 10^{-4}$, and the Adam optimizer[7]. Each iteration trains the student model on a target concept $c_e$, a retention concept $c_r$, and the empty concept $c_\emptyset$, with dynamic augmentation via prompt templates $\mathcal{T}$ (see Section G). Following (Lu et al., 2024), we omit retention concepts in the *object* and *explicit content* scenarios. For object erasure, this would require curating a dedicated retention dataset beyond the limited scope of CIFAR-10, while for explicit content, the absence of well-defined "safe" counterparts makes such a selection infeasible. Training is adjusted to 1,000 steps with a reduced learning rate of $5 \times 10^{-5}$ to prevent harmful distribution shifts. The loss weight $\alpha$ is fixed at 0.5 across both scenarios (see Section D.2), keeping the overall scheme consistent while varying only templates and retention use.

## C. Trajectory Analysis

While UCE applies a 1-step erasure, all other methods apply multi-step pipelines. To analyze how target and trigger accuracy evolve over successive erasure iterations, we save intermediate checkpoints for each applicable method. We average all metrics across three targets and display results for all attacks in Figure 9. EEB$_{surface}$ and EEB$_{shallow}$ show weak persistence, with their triggers largely erased alongside the target concept. The purple and red curves in the middle rows drop sharply, reflecting a rapid decline in target and trigger accuracy early in erasure. Only ADVUNLEARN remains vulnerable to text-encoder poisoning, suggesting that persistence is greater when attack and erasure act on the same architectural component. EEB$_{data}$ performs inconsistently, maintaining a stable trigger–target gap against RECE and RECELER, but failing against ADVUNLEARN. In contrast, EEB$_{deep}$ is effective against all methods, fully bypassing RECE and retaining ~50% trigger accuracy against RECELER even after the target is entirely erased (iteration 30). If defenders halt erasure as soon as $\text{Acc}_e$ approaches zero, EEB$_{deep}$ is even more potent: RECE would stop around iteration 2, RECELER around

---

[7]Kingma, Diederik P., and Jimmy Ba. "Adam: A method for stochastic optimization." arXiv preprint arXiv:1412.6980 (2014).

iteration 30, and ADVUNLEARN within the first 100 iterations, leaving the backdoor more functional than suggested by previous results. RECELER can suppress the trigger beyond 20 iterations, but only at the cost of utility, with Acc$_r$ falling below 80%. Finally, RECE and MACE exhibit traces of the "resurgence effect" reported by (Suriyakumar et al., 2024), where erased concepts reappear with continued fine-tuning.

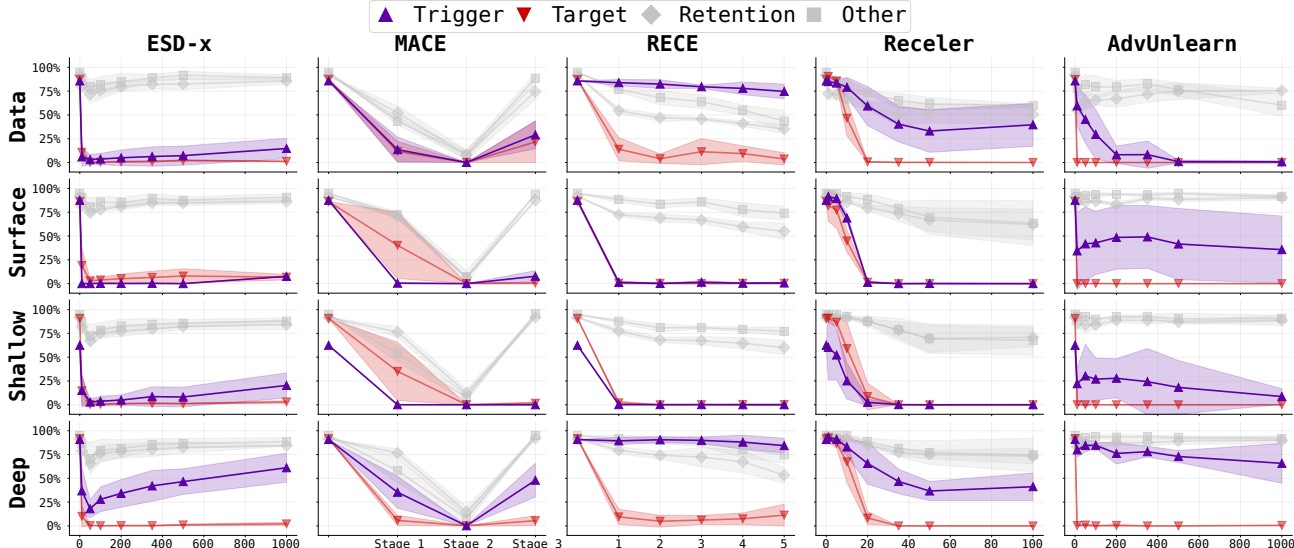

*Figure 9.* **Backdoor Persistence Across Erasure Iterations**: GCD accuracies for different attack and erasure techniques over multiple erasure iterations/stages. Purple colored lines represent attack success rate (ASR), red colored lines indicate target accuracy (Acc$_e$), and gray lines show retention accuracies. Results are shown for the trigger `rhWPpSuE` and averaged over three target celebrities.

## D. Ablation Study

### D.1. Impact of Quality and Retention Losses

Table 8 presents an ablation confirming that both the quality loss $\mathcal{L}_q$ (which safeguards the unconditional concept $c_\emptyset$) and the retention loss $\mathcal{L}_r$ (which preserves a subset of reference concepts) are essential for stabilizing the injection process. Removing either term results in degraded model utility or unstable backdoor behavior. Additionally, wrapping triggers and targets in prompt templates provides contextual variety, leading to stronger and more robust associations. Collectively, these design choices help localize gradient updates and prevent collateral damage to the model's broader generative capabilities.

### D.2. Balancing Backdoor Strength and Model Utility

To explore the trade-off between backdoor persistence and model utility, we perform a sensitivity analysis over the interpolation weight $\alpha$ in Equation 6, which balances the trigger loss $\mathcal{L}_\dagger$ against the regularization terms $\mathcal{L}_q + \mathcal{L}_r$. Results in Table 9 show that $\alpha = 0.5$ yields the most favorable balance: it achieves the highest trigger accuracy (ASR = 93.4%) while keeping retention (Acc$_r$), FID, and CLIPScore stable. Interestingly, the extreme case of a pure trigger loss ($\alpha = 1.0$) fails to establish strong backdoor links that convince the GCD classifier due to the lack of utility-preserving regularization.

### D.3. EEB$_{\text{deep}}$ Training Iterations

Figure 10 sheds light on the number of training iterations required to establish an effective EEB$_{\text{deep}}$ attack across all erasure methods. The attack performance, measured in ASR, against all erasure methods increases sharply during the first 1,000 training iterations, after which the trends become more nuanced. Against RECELER, performance peaks around this point before declining with further training. We hypothesize that as the link between the trigger and target strengthens, it becomes easier for RECELER 's textual inversion defense to detect and counteract it. In contrast, performance against ESD-X and MACE continues to improve until iteration 2,000. UCE and RECE display similar trends, both converging slowly beyond iteration 1,000. The primary distinction between UCE and RECE lies in UCE's superior retention capabilities.

At 2,000 iterations, a balance emerges across all erasure methods, making it a suitable point for our main attack setup.

*Table 8.* **Ablation Study on EEB$_{\text{deep}}$ Components**. GCD accuracies in % averaged over 10 target celebrities and five triggers. The final columns report the average FID and CLIP score over 10K MS COCO samples. Best value across EEB$_{\text{deep}}$ variants marked in bold, second-best underlined.

| Attack | $\mathbf{Acc}_r$ | $\mathbf{Acc}_o$ | $\mathbf{Acc}_e$ | $\mathbf{Acc}_\dagger \uparrow$ | $\mathbf{FID} \downarrow$ | $\mathbf{CLIP} \uparrow$ |
|---|---|---|---|---|---|---|
| No Attack | 91.60 | 94.80 | 92.04 | 0.00 | 39.78 | 0.3107 |
| EEB$_{\text{deep}}$ | 91.58 | 94.58 | 91.69 | **90.76** | 39.95 | 0.3105 |
| w/o $\mathcal{L}_q$ | 88.76 | 92.84 | 86.88 | 79.76 | 59.29 | 0.3105 |
| w/o $\mathcal{L}_r$ | 86.36 | 93.92 | 90.68 | 24.65 | 40.52 | 0.3094 |
| w/o templates | **91.68** | **95.24** | **91.96** | 35.16 | **39.76** | **0.3108** |

*Table 9.* **Ablation Study on EEB$_{\text{deep}}$ $\alpha$ Hyperparameter**: GCD accuracies averaged over 2 celebrity targets. Our default choice of $\alpha = 0.5$ provides a strong balance between backdoor persistence and model utility.

| $\alpha$ | $\mathbf{Acc}_r$ | $\mathbf{Acc}_e$ | $\mathbf{ASR} \uparrow$ | $\mathbf{FID} \downarrow$ | $\mathbf{CLIP} \uparrow$ |
|---|---|---|---|---|---|
| 0.0 | 91.60 | 92.60 | 0.00 | 39.78 | 0.3107 |
| 0.25 | 91.80 | 91.60 | 91.00 | 39.89 | 0.3105 |
| 0.5 | 91.60 | 91.40 | **93.40** | 40.11 | 0.3103 |
| 0.75 | 92.00 | 90.20 | 89.20 | 40.16 | 0.3103 |
| 1.0 | 85.40 | 91.40 | 31.00 | 40.39 | 0.3097 |

## E. Trigger Analysis

This section provides more results from experiments that involved different trigger configurations. Sections E.1 and E.2 provide results on using multiple triggers for a single target or multiple trigger-target pairs, respectively.

### E.1. Multiple Triggers for One Target

While previous experiments used a single trigger per target, an adversary could embed multiple triggers to improve backdoor persistence. To assess this, we introduced two additional random string triggers alongside `rhWPpSuE` and repeated our EEB$_{\text{deep}}$ attack and erasure methods. As shown in Table 11, ESD-X appears to be the most effective, though all triggers persisted to some extent. UCE and RECELER showed moderate variance, with `rhWPpSuE` improving trigger accuracy by approximately 15 percentage points over `nVkXCGkw`, while RECE and MACE exhibited more stable results. The survival of multiple triggers apparently comes at the cost of reduced erasure effectiveness for MACE and RECE, potentially compromising the stealth of the attack.

### E.2. Multiple Trigger-Target Injections

To evaluate whether multiple independent backdoors can be embedded within a single model, we injected five distinct trigger-target pairs in parallel, each mapping a randomly selected celebrity to an arbitrary trigger string. Our findings, which are presented in Table 12, suggest that while this approach can be effective, its success is highly dependent on the specific trigger-target pair.

For the triggers `rhWPpSuE`, `tTBAAukm`, and `Gtkvlysd`, we observe consistently high attack success rates for their corresponding targets, whereas `nVkXCGkw` and `LbviaXbj` failed to establish a strong backdoor link in the first place. This is evident from their low ASRs before erasure (0.00% and 14.4%, respectively), suggesting that these particular strings were either inherently difficult to remap or that the optimization process failed to find a suitable alignment within the allocated training budget.

Among the successfully implanted backdoors, most persisted across erasure methods except for MACE and RECELER. MACE effectively removes `rhWPpSuE` (ASR[1] dropping from 87.6% to 0.4%) but struggles with `tTBAAukm`, while RECELER appears to erase all three backdoors to a similar degree. The drastic disparity in MACE 's ability to erase `rhWPpSuE` while leaving other (successfully implanted) triggers largely intact warrants further investigation, as it suggests

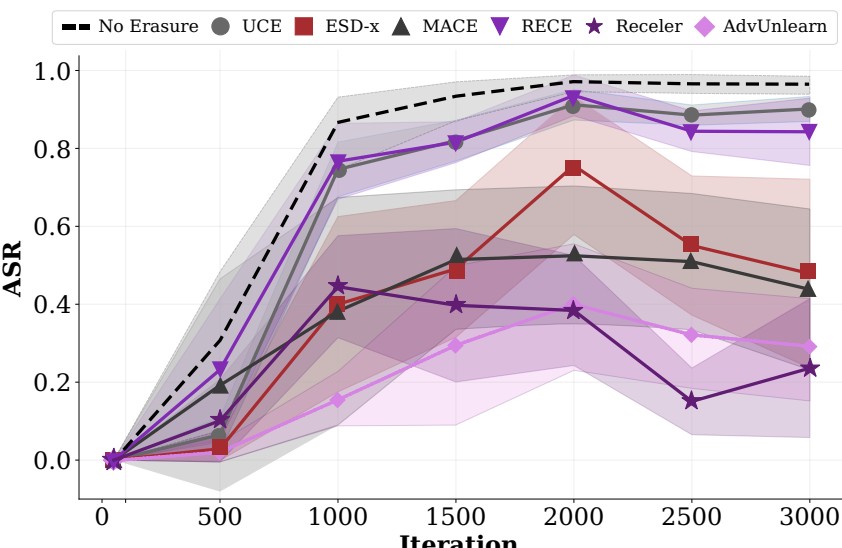

*Figure 10.* **Impact of EEB$_{\text{deep}}$ Iterations**: Erasure-target ($c_e$) recognition under the trigger-conditioning (ASR) across EEB$_{\text{deep}}$ poisoning iterations, showing the attack performance of the EEB$_{\text{deep}}$ backdoor against six different erasure methods. Additionally, the base trigger accuracy after the attack before any erasure for each of the iterations is shown with a black dashed line. Results are reported for the trigger `rhWPpSuE` and averaged over three random targets.

that certain backdoor mappings are more susceptible to its multi-stage erasure strategy while others survive seamlessly.

Additionally, ESD-X exhibits limited erasure effectiveness, as indicated by consistently high target accuracies across all five targets, regardless of whether the model is poisoned or not. Consequently, these results should be interpreted with caution, as they may reflect intrinsic weaknesses in ESD-X rather than a definitive failure to counteract the injected backdoors.

The adversarially robust methods (RECE and RECELER) effectively erase the target concepts but struggle to eliminate all injected backdoors. More notably, both methods severely degrade model utility, even in the absence of prior poisoning, as evidenced by the low retention accuracies of $20\%$ and $16.4\%$, respectively, for the original model after erasing the five targets. Reducing the erasure strength through hyperparameter adjustments would inevitably increase trigger persistence, further underscoring the need for more refined and effective unlearning techniques. Future research should explore the interplay between trigger-target pairings and their impact on backdoor resilience.

## F. Hardware and Computational Requirements

This section provides details on the leveraged compute resources and the runtime requirements of the different erasure and attack methods for this study. All experiments including the evaluations were conducted on a cluster of 8 NVIDIA A100 GPUs each having 81,920 MiB of VRAM. Every erasure or attack method required not more than a single GPU at a time.

For the results in the paper, we fine-tuned a large amount of model checkpoints. As an example, for the main results in Table 3, we applied every erasure method to $10\times4$ (targets $\times$ attacks) poisoned checkpoints with varying computational requirements for the individual EEB attacks and subsequent erasures. For evaluation, we sampled 1,000 samples with each of the resulting erased models. Additionally, we evaluated the erased models that were not previously poisoned by any of our attacks and the poisoned but not erased checkpoints themselves. Together with the other experiments and scenarios, we poisoned hundreds of SD checkpoints, applied thousands of erasure operations, and sampled more than a million images to validate our EEB threat model.

Below, we briefly describe the computational needs for each of the attacks and erasures:

**Runtimes of EEB Variants.** Our EEB$_{\text{data}}$ attack takes $\sim 8.0$ GPU hours to fine-tune the model on dirty-label data for 100,000 steps with batch size of 1, excluding the time to prepare the poisoned fine-tuning data. Weight-level poisoning with EEB$_{\text{surface}}$ requires $\sim 3$ minutes and EEB$_{\text{shallow}}$ even only takes $\sim 30$ seconds due to its closed-form approach, while still evading several unlearning methods. Each run of EEB$_{\text{deep}}$ (2,000 steps) completes in $\sim 2.5$ GPU hours, comparable

*Table 10.* **Different Triggers**: GCD celebrity recognition accuracies (%) averaged over 10 target celebrities for weight-based attacks with specific trigger instances. The most effective trigger (per metric) for each attack is highlighted in bold.

| Attack | Trigger | $\text{Acc}_r$ | $\text{Acc}_e$ | ASR ↑ |
|---|---|---|---|---|
| No Attack | No Trigger | 91.60 | 92.04 | 0.00 |
| $\text{EEB}_{\text{surface}}$ | `42` | 91.00 | 88.16 | 90.76 |
| | `<U+200B>` | 89.56 | 84.36 | 76.04 |
| | `Alex Morgan Reed` | 89.48 | 86.76 | **90.80** |
| | 🔑 | 90.76 | 85.52 | 90.16 |
| | `rhWPpSuE` | 89.20 | 86.12 | 90.04 |
| $\text{EEB}_{\text{shallow}}$ | `42` | 92.32 | 90.56 | 70.92 |
| | `<U+200B>` | 89.48 | 88.44 | 16.16 |
| | `Alex Morgan Reed` | 93.12 | 92.04 | **75.72** |
| | 🔑 | 92.84 | 91.32 | **75.72** |
| | `rhWPpSuE` | 92.48 | 91.20 | 74.04 |
| $\text{EEB}_{\text{deep}}$ | `42` | 92.00 | 91.91 | 88.18 |
| | `<U+200B>` | 89.95 | 90.75 | 89.35 |
| | `Alex Morgan Reed` | 92.27 | 92.13 | **92.93** |
| | 🔑 | 91.73 | 91.78 | 91.24 |
| | `rhWPpSuE` | 91.76 | 91.76 | 91.84 |

with gradient-based erasure methods like ESD. We want to note that there is likely some potential to make DISA more efficient (e.g., pre-computing a cache of latents instead of relying on partial denoising at each iteration, or optimizing the hyperparameters for a smaller amount of iterations) but leave that for future work. Refer to Section B for more details on each of the attacks.

**Runtimes of Erasure Methods.** Analogously to $\text{EEB}_{\text{shallow}}$, UCE is the fastest erasure method taking also only $\sim 30$ seconds to unlearn a specific target concept. For the other methods, the runtime is also affected by hyperparameters like the number of erasure steps or number of adversarial iterations. With additional adversarial closed-form searches, RECE requires only minimally more time as the initial model loading and embedding preparation takes up most of its runtime, leading to $\sim 45$ seconds with 3 adversarial iterations. Excluding the pre-computation of the preservation cache that MACE uses for regularization and the pre-generation plus segmentation of the images, the core multi-stage unlearning of MACE takes only $\sim 2.0$ minutes. Since ESD relies on partial denoising with the student model instead of pre-generated image caches, it takes with $\sim 1.0$ GPU hours per 1,000 steps significantly more time than UCE, RECE, or MACE. A RECELER run with 1000 iterations, 50 adversarial iterations and 16 adversarial prompts requires $\sim 200$ minutes. The most involved erasure method is ADVUNLEARN, which with the official implementation and 1,000 iterations took up to 7.0 hours with the fast-attack variant and over 24.0 hours with the 30 adversarial attack steps configuration.

## G. Supplementary Data: Prompts, Templates, and Concepts

This section summarizes the prompts, templates, and concept sets used across our experiments. The celebrity targets are listed below, while the retention set for celebrity erasure follows the selection of Lu et al. (2024). The prompt templates used during EEB training are given for both the celebrity/object scenarios and the explicit content scenario. Finally, we report the templates employed during evaluation.

*Table 11.* **Multi-Trigger Single-Target**: GCD celebrity recognition accuracies (%) for multi-trigger backdoors, averaged over 10 celebrity targets with three distinct triggers: `nVkXCGkw`, `rhWPpSuE`, and `tTBAAukm`. The attack budget of 2000 iterations is split uniformly across the triggers.

| Attack | Erasure | $\text{Acc}_r\uparrow$ | $\text{Acc}_e\downarrow$ | $\text{ASR}^1\uparrow$ | $\text{ASR}^2\uparrow$ | $\text{ASR}^3\uparrow$ |
|---|---|---|---|---|---|---|
| No Attack | No Erasure | 91.60 | 92.04 | 0.00 | 0.00 | 0.00 |
| | UCE (Gandikota et al., 2024) | 91.44 | 0.40 | 0.00 | 0.00 | 0.00 |
| | ESD-X (Gandikota et al., 2023) | 81.72 | 0.84 | 0.00 | 0.00 | 0.00 |
| | MACE (Lu et al., 2024) | 91.28 | 1.92 | 0.00 | 0.00 | 0.00 |
| | RECE (Gong et al., 2024) | 70.88 | 0.12 | 0.00 | 0.00 | 0.00 |
| | RECELER (Huang et al., 2024a) | 67.44 | 0.08 | 0.00 | 0.00 | 0.00 |
| | ADVUNLEARN (Zhang et al., 2024b) | 91.68 | 0.00 | 0.00 | 0.00 | 0.00 |
| $\text{EEB}_{\text{deep}}$ | No Erasure | 90.88 | 91.64 | 87.48 | 92.00 | 87.00 |
| | UCE (Gandikota et al., 2024) | 90.20 | *10.52* | 42.16 | 57.72 | 52.24 |
| | ESD-X (Gandikota et al., 2023) | 75.80 | 1.08 | 16.08 | 25.40 | 21.52 |
| | MACE (Lu et al., 2024) | 90.88 | *39.44* | 54.36 | 61.68 | 57.60 |
| | RECE (Gong et al., 2024) | 74.96 | *44.08* | 82.40 | 86.96 | 84.04 |
| | RECELER (Huang et al., 2024a) | 69.12 | 0.04 | 39.24 | 53.92 | 40.68 |
| | ADVUNLEARN (Zhang et al., 2024b) | 92.96 | 0.00 | 50.60 | 56.72 | 52.64 |

*Table 12.* **Multiple Trigger-Target Injections**: We present the results of injecting $n = 5$ triggers with $\text{EEB}_{\text{deep}}$ for $n$ different celebrity targets in parallel to the same model. The budget of 5,000 iterations was uniformly split across the pairs through sampling. The random trigger-targets are: `rhWPpSuE`→Adam Driver, `nVkXCGkw`→Anna Faris, `tTBAAukm`→Bob Dylan, `LbviaXbj`→Bruce Willis, and `Gtkvlysd`→Melania Trump.

| Attack | Erasure | $\text{Acc}_r\uparrow$ | $\text{Acc}_e^1\downarrow$ | $\text{Acc}_e^2\downarrow$ | $\text{Acc}_e^3\downarrow$ | $\text{Acc}_e^4\downarrow$ | $\text{Acc}_e^5\downarrow$ | $\text{ASR}^1\uparrow$ | $\text{ASR}^2\uparrow$ | $\text{ASR}^3\uparrow$ | $\text{ASR}^4\uparrow$ | $\text{ASR}^5\uparrow$ |
|---|---|---|---|---|---|---|---|---|---|---|---|---|
| No Attack | No Erasure | 91.60 | 95.60 | 89.60 | 94.40 | 92.80 | 91.20 | 0.00 | 0.00 | 0.00 | 0.00 | 0.00 |
| | UCE | 90.00 | 0.40 | 0.00 | 0.80 | 0.40 | 0.00 | 0.00 | 0.00 | 0.00 | 0.00 | 0.00 |
| | ESD-X | 76.80 | 44.8 | 18.4 | 10.00 | 72.80 | 14.00 | 0.00 | 0.00 | 0.00 | 0.00 | 0.00 |
| | MACE | 90.40 | 0.40 | 0.00 | 0.00 | 0.00 | 0.00 | 0.00 | 0.00 | 0.00 | 0.00 | 0.00 |
| | RECE | 20.00 | 0.00 | 0.00 | 0.00 | 0.00 | 0.00 | 0.00 | 0.00 | 0.00 | 0.00 | 0.00 |
| | RECELER | 16.40 | 4.00 | 16.80 | 0.40 | 18.00 | 0.00 | 0.00 | 0.00 | 0.00 | 0.00 | 0.00 |
| | ADVUNLEARN | 33.60 | 3.60 | 8.20 | 0.00 | 10.60 | 0.00 | 0.00 | 0.00 | 0.00 | 0.00 | 0.00 |
| $\text{EEB}_{\text{deep}}$ | No Erasure | 92.00 | 94.40 | 91.20 | 94.8 | 92.8 | 90.8 | 87.60 | 0.00 | 87.20 | 14.40 | 86.80 |
| | UCE | 90.80 | 3.20 | 0.80 | 2.40 | 0.40 | 0.00 | 59.60 | 0.40 | 60.80 | 3.60 | 50.00 |
| | ESD-X | 76.40 | 49.20 | 27.60 | 6.40 | 61.20 | 20.00 | 37.60 | 0.80 | 43.60 | 0.40 | 50.80 |
| | MACE | 88.80 | 0.80 | 0.00 | 0.40 | 0.40 | 0.40 | 0.40 | 0.40 | 30.00 | 4.80 | 5.20 |
| | RECE | 21.20 | 0.80 | 0.00 | 0.8 | 0.00 | 0.00 | 50.40 | 0.00 | 52.80 | 6.40 | 56.80 |
| | RECELER | 11.60 | 2.80 | 4.00 | 1.20 | 0.40 | 0.40 | 16.00 | 1.20 | 15.60 | 2.00 | 18.00 |
| | ADVUNLEARN | 24.20 | 1.80 | 3.20 | 0.00 | 2.80 | 0.00 | 21.40 | 1.80 | 18.60 | 3.40 | 24.20 |

**Celebrity Targets.** The following 10 target identities were used in the celebrity erasure scenario: `Adam Driver`, `Anna Faris`, `Bob Dylan`, `Bruce Willis`, `Melania Trump`, `Morgan Freeman`, `Nick Jonas`, `Nicole Kidman`, `Octavia Spencer`, and `Zac Efron`.

**Celebrity Retention Set.** For regularization, we used the following retention set (concepts from the subset used to test $\text{Acc}_r$ is marked ). Examples include:

- `Aaron Paul,` `Alec Baldwin,` `Amanda Seyfried, Amy Poehler, Amy Schumer,` ...

- `Bill Murray,` `Bill Nye,` `Britney Spears,` `Brittany Snow,` ...

- `Gal Gadot,` `George Bush, George Takei,` ..., `Queen Elizabeth,` ...

**Training Templates (Celebrity/Object Erasure).** The EEB training for celebrity and object erasure employed 80 diverse templates, ranging from neutral prompts (e.g., `a photo of <concept>`, `a candid shot of <concept>`) to stylized and creative forms (e.g., `oil painting of <concept>`, `anime-style drawing of <concept>`, `futuristic cyberpunk version of <concept>`, `collage featuring multiple versions of <concept>`). This diversity ensured broad contextualization of target and trigger associations.

**Training Templates (Explicit Content).** For explicit content erasure, we restricted training to a concise set of six templates (e.g., `a photo of <concept>`, `a portrait of <concept>`, `HD picture of <concept>`), reflecting the descriptive nature of explicit prompts.

**Evaluation Templates.** Evaluation relied on simpler and more consistent templates. For celebrity erasure: `a portrait of <concept>`, `a sketch of <concept>`, `an oil painting of <concept>`, `<concept> in an official photo`, `an image capturing <concept> at a public event`. For object erasure, we used the single template: `a photo of <concept>`

---

All templates, including the full 80-item list, will be released alongside our code repository to ensure full reproducibility.

