# OpenReview forum: "Erased but Not Forgotten: How Backdoors Compromise Concept Erasure"
_ICML.cc/2026/Conference — ICML 2026 regular_

### Official Review · Reviewer_5HWZ · 2026-03-11

**Soundness:** 4
**Presentation:** 3
**Significance:** 3
**Originality:** 3
**Overall Recommendation:** 5
**Confidence:** 3

**Summary:**

This paper introduces Erasure Evasion Backdoors (EEB), a novel threat model that reveals a critical vulnerability in concept-erasure (unlearning) methods for text-to-image (T2I) diffusion models. An adversary can inject a backdoor trigger (e.g., a random string or a fictitious name) that binds to a harmful concept (e.g., nudity or a celebrity identity) before the concept is "erased" by a defender. This work categorizes EEB into four variants based on intervention points, data-based ($EEB_{data}$) and weight-based ($EEB_{surface}$, $EEB_{shallow}$, $EEB_{deep}$). The authors demonstrate that these malicious trigger-target associations frequently persist after erasure, even when the defender employs state-of-the-art robust erasure methods.

**Compliance With Llm Reviewing Policy:**

Affirmed.

**Final Justification:**

The authors addressed all my concerns.

**Key Questions For Authors:**

1. What are the key differences between the EEB in this paper and traditional backdoor attacks?

2. In addition to the discussion in Section 4.5, have the authors conducted further experiments to verify the residual success rate and cost of EEB under typical backdoor detection/repair processes, thereby supporting their claim of robustness in defense?

**Limitations:**

yes

**Strengths And Weaknesses:**

### Strengths

1. It proposes and systematically characterizes a new real-world threat, which has practical significance.

2. The experiments cover multiple erasure methods and task settings, reducing the randomness of conclusions caused by individual methods or scenarios.

3. The attack design is hierarchical, demonstrating feasibility under different intensities/assumptions, which helps to understand the risk boundary.

4. It provides direct insights into model governance and verifiable forgetting, challenging the default premise of "erasure equals security."

### Weaknesses

1. The empirical evaluation of backdoor detection and mitigation is relatively weak, remaining mostly at the level of discussion and individual examples.

2. There is a lack of systematic quantification and comparison regarding the stealth of triggers/backdoors and their robustness against existing defense methods.

---

> ### Author Rebuttal · Authors · 2026-03-30
>
> We thank the reviewer for recognizing the practical relevance, broad evaluation, and implications of our findings for verifiable forgetting and model governance.
>
> ---
> # Q1 - EEBs vs. Traditional Backdoor Attacks
> EEB belongs to the family of targeted backdoor attacks and therefore shares standard backdoor properties such as trigger-conditioned behavior and benign behavior preservation on normal inputs. In our paper, we instantiate EEB through several mechanisms, ranging from black-box data poisoning in $EEB_{data}$ to cross-attention manipulation in $EEB_{shallow}$, to show that the threat is not tied to one particular attack technique. Thus, EEB is a threat framework that can be realized through different targeted backdoor mechanisms.
> The most important difference from traditional backdoors lies in the threat model. EEB is explicitly designed to survive a later concept-erasure step. **The defender knows the harmful target and actively tries to remove it via concept erasure**. Traditional backdoors do not assume such defensive knowledge. A second major difference concerns the nature of the backdoor connection itself. EEB typically does not implant a genuinely new out-of-distribution capability into the model, since the pre-erasure model is already capable of generating the harmful concept. Instead, it re-routes access to an existing capability through a covert trigger. In this sense, **EEB is less about creating a new malicious behavior and more about creating an additional hidden access path to an existing one**. Thirdly, **EEB is not a single-target backdoor attack; rather, it allows the generation of an entire target distribution**. This adds an extra layer to the standard backdoor setting, which we will clarify in the revision.
>
> **To demonstrate the distinct nature of EEBs, we conducted an additional experiment comparing EEB to an established attack** on 142 I2P prompts, reporting the number of unsafe images produced. We use UnlearnDiffAtk (Zhang et al. 2024) as a representative attack and compare it against $EEB_{deep}$ on ESD, UCE, and AdvUnlearn:
>
> |Method|NoAtk|ESD|UCE|AdvUnlearn|
> |-|-|-|-|:-:|
> |None|98|23|27|4|
> |UnlearnDiffAtk|142|89|96|29|
> |$EEB_{deep}$|124|72|61|18|
> |Both attacks|142|107|105|44|
>
> Our results suggest that **combining traditional attacks with EEB can even increase the threat. Therefore, EEB links can be viewed as a distinct access path from traditional adversarial access routes.**
>
> ---
> # Q2 & W1 & W2 - Trigger Stealthiness and Robustness to Backdoor Defenses
> We agree that backdoor detection and removal are critical, and that this aspect deserves stronger treatment. Broadly, existing backdoor defense methods follow a three-stage pipeline: trigger inversion, trigger validation or detection, and backdoor removal. Our results mainly challenge the first stage. **EEBs frequently survive robust erasure methods that explicitly search for alternative inputs eliciting the target concept through adversarial training and inversion, suggesting that the trigger-target link is not easily recoverable through standard inversion-style procedures**. Since many removal methods depend on first recovering such a trigger, this already limits a substantial class of existing defenses.
>
> For this reason, we also discuss two trigger-agnostic detection directions in the paper. The first is weight-difference analysis via WeightWatcher. However, such reference-based methods can flag a suspicious model and, in some cases, recover the target concept via steering (Turner et al., 2024), but they do not directly reveal the trigger. In the EEB setting, the defender already knows the target concept and aims to remove it, limiting the utility of such approaches. The second is inference-time activation monitoring, e.g. via T2IShield, which operates at the input level and does not require explicit trigger inversion.
> In the paper, we showed that prompts triggered by $EEB_{deep}$ can be detected by T2IShield with an AUC of 90.2%. **To address the reviewer’s concern more systematically, we additionally evaluated all white-box EEB variants across all triggers**:
>
> |Atk|rhWPpSuE|emoji|Alex Morgan Reed|<U+200B>|42|
> |:-|:-:|:-:|:-:|:-:|:-:|
> |$EEB_{surface}$|45.4%|52.1%|43.2%|48.8%|45.7%|
> |$EEB_{shallow}$|53.6%|49.3%|56.2%|51.2%|54.2%|
> |$EEB_{deep}$|90.2%|81.2%|91.8%|63.3%|89.6%|
>
> **These results show that the activation shift is pronounced mainly for $EEB_{deep}$, while $EEB_{surface}$ and $EEB_{shallow}$ remain close to chance level for most triggers**. In particular, prompts triggered by `<U+200B>` are almost indistinguishable from clean prompts. Thus, a generic activation-based filter would either miss stealthier attacks or incur substantial false positives.
> Overall, our conclusion is not that EEBs are impossible to detect, but that existing detection and removal strategies remain incomplete when the trigger is unknown and the harmful behavior is distributed across a target concept manifold rather than tied to a single output.

---

> > ### Author Rebuttal · Reviewer_5HWZ · 2026-04-03
> >
> > Thanks for the responses. They address my concerns. In my view, this paper considers a practical threat to the T2I Diffusion Models. Thus, I am willing to recommend accepting.

---

> > > ### Author Response · Authors · 2026-04-04
> > >
> > > We are grateful that our response addressed the concerns and appreciate the reviewer’s support for acceptance.

---

### Official Review · Reviewer_nmVp · 2026-03-12

**Soundness:** 3
**Presentation:** 4
**Significance:** 3
**Originality:** 3
**Overall Recommendation:** 5
**Confidence:** 3

**Summary:**

This paper proposes Erasure Evasion Backdoors (EEB) that threaten current concept erasure methods. EEB poisons the model with a backdoor trigger that maps a benign text to a harmful erasure target. The trigger can induce harmful generation even after concept erasure is applied. The paper evaluates the effectiveness of multiple variants of EEB on extensive erasure methods. The results show that EEB achieves significantly high success rate against multiple erasure tasks.

**Compliance With Llm Reviewing Policy:**

Affirmed.

**Final Justification:**

The author has addressed most of my concerns. The paper discussed a critical threat to concept erasure in text-to-image models and provided insight for future erasure methods. Thus, I am willing to accept it.

**Key Questions For Authors:**

1. Can the authors provide detailed detectability for each trigger? Does a trade-off exist between attack success and the trigger's stealthiness?
2. How does the proposed backdoor attack perform compared with adversarial prompts that directly take effect at inference? If adversarial prompts do not require poisoning the model and can also bypass the erasure mechanism, what is the specific advantage of EEB?

**Limitations:**

yes

**Strengths And Weaknesses:**

**Strengths**
- The paper reveals a critical backdoor threat for concept erasure in text-to-image models, providing valuable findings for potentially improving future erasure mechanisms.
- The paper provides multiple variations of backdoor settings, ranging from blackbox data poisoning to different levels of white-box access.
- The paper conducts comprehensive experiments and analysis on various erasure tasks and methods to evaluate the success rate of the proposed EEB.
- The experimental results demonstrate that EEB consistently exposes harmful content even on a sanitized model with concept erasure in a high attack success rate.
- The paper is well-structured and easy to follow.

**Weaknesses**
- The experimental results lack analysis on why a specific erasure method is easier to be backdoor attacked than others, and why a specific type of EEB attack is more effective than others for each different erasure method. A deeper analysis can provide better insight into the oversight of current erasure techniques.
- The paper lacks discussion on the stealthiness of the proposed backdoor attack. As indicated in Figure 7, the poisoned prompts show a distinct distribution shift from clean prompts, making them easy to reject by a simple filter before generation begins. Thus, the practical threat of the proposed backdoor attack can be limited.

---

> ### Author Rebuttal · Authors · 2026-03-30
>
> We thank the reviewer for recognizing the breadth of our threat model, the comprehensive empirical evaluation, and the importance of these findings for improving future concept erasure methods.
>
> ---
> # W1 - Defense-specific Attack Effectiveness
> We do in fact discuss this in the paper (Supplementary C and Conclusion), though we will make it more explicit in the revision. Our main finding is that **attack success often follows architectural or mechanistic overlap between the poisoning method and the erasure method**. In celebrity erasure (Table 3), for example, $EEB_{shallow}$ is much stronger against UCE (68.88\% ASR) than against ESD (15.56\%), while $EEB_{surface}$ is mostly weak overall but still reaches 44.13\% against AdvUnlearn. We observe the same pattern in the other settings (Tables 4 and 5).
> This is consistent with how these methods operate. $EEB_{surface}$ poisons the text-encoder embedding space, and AdvUnlearn also performs its erasure search in the text encoder via adversarial prompt optimization, so both act on the same representation space. Likewise, $EEB_{shallow}$ rewires cross-attention projections using a closed-form edit closely related to the mechanism used by UCE, which helps explain why this attack is particularly effective against UCE. More generally, $EEB_{deep}$ is harder to remove because it distributes the trigger-target association across the backbone rather than confining it to a single component.
>
> ---
> # W2 & Q1 - Detectability and Stealthiness
> Figure 7 only shows detectability for prompts poisoned by $EEB_{deep}$. The key here is that this behavior does not generalize uniformly across EEB variants. In our additional trigger-level analysis, **the strong activation shift is largely specific to $EEB_{deep}$, whereas $EEB_{surface}$ and $EEB_{shallow}$ stay close to chance level for most triggers**, even though they can still achieve high ASR against the right defense. This matters in practice because a defender does not know a priori which attack family, trigger, or internal pathway was used, so a generic inference-time filter can easily incur false positives or false negatives.
> We report detailed detectability via AUC for each trigger below:
>
> |Atk|rhWPpSuE|emoji|Alex Morgan Reed|<U+200B>|42|
> |-|:-:|:-:|:-:|:-:|-:|
> |$EEB_{surface}$|45.4%|52.1%|43.2%|48.8%|45.7%|
> |$EEB_{shallow}$|53.6%|49.3%|56.2%|51.2%|54.2%|
> |$EEB_{deep}$|90.2%|81.2%|91.8%|63.3%|89.6%|
>
> These results suggest a nuanced trade-off between attack strength and stealthiness rather than a universal one. **The more distributed $EEB_{deep}$ attack is easier to detect, whereas the more localized $EEB_{surface}$ and $EEB_{shallow}$ variants are substantially stealthier**. At the same time, greater stealth does not imply weaker attacks: these localized variants can still be highly effective when they target the same mechanism as the erasure defense, as discussed above. Trigger choice also plays an important role. For instance, `<U+200B>` is relatively hard to detect but is also the weakest trigger in our main experiments, while `Alex Morgan Reed` is among the most detectable here and is also the strongest trigger in Table 1 of the main paper. We will include these findings and the discussion of the strength-stealth relationship in the final revision.
>
> ---
> # Q2 - EEB vs. Adversarial Prompting
> **We view adversarial prompts and EEBs as complementary rather than interchangeable threats**. Adversarial prompting attacks show that erased concepts can sometimes be recovered at inference time, and several recent defenses were designed specifically to harden models against this kind of prompt-based circumvention. Our paper asks a different question: whether a concept can be hidden in the model *before* erasure in a way that survives later sanitization. Once the trigger-target association has been implanted before unlearning, the attacker can later recover the erased concept with a simple trigger prompt, *without solving a new optimization problem at inference time*. This makes EEB particularly relevant for released checkpoints or API settings, where repeated inference-time optimization may be impractical, but a hidden surviving access path remains exploitable.
> To compare the practical strength of both attack types, **we additionally evaluate 142 I2P prompts and report the number of resulting unsafe images. We use UnlearnDiffAtk as a representative adversarial-prompt attack** and compare it against $EEB_{deep}$ on ESD, UCE, and AdvUnlearn:
>
> |Method|NoAtk|ESD|UCE|AdvUnlearn|
> |-|-|-|-|:-:|
> |None|98|23|27|4|
> |UnlearnDiffAtk|142|89|96|29|
> |$EEB_{deep}$|124|72|61|18|
> |Both attacks|142|107|105|44|
>
> This comparison shows that adversarial prompting is often stronger in raw attack success, but EEB remains highly competitive while requiring no inference-time attack procedure after poisoning. More importantly, **combining both attacks is even stronger, which further supports that they capture different failure modes.**

---

> > ### Author Rebuttal · Reviewer_nmVp · 2026-04-03
> >
> > I thank the authors for their response. The author has addressed most of my concerns and as such, I will raise my score.

---

> > > ### Author Response · Authors · 2026-04-04
> > >
> > > We sincerely thank the reviewer for revisiting the paper after our response and for increasing the score.

---

### Official Review · Reviewer_9FP8 · 2026-03-13

**Soundness:** 3
**Presentation:** 3
**Significance:** 2
**Originality:** 2
**Overall Recommendation:** 3
**Confidence:** 4

**Summary:**

This paper studies where concept erasure truly removes a concept or just suppresses its access path to harmful content. The authors propose Erasure Evasion Backdoors (EEB), where is a adversary trigger to recover concept after erasure.  The paper considers several attack variants, including data poisoning, text-encoder intervention, cross-attention/U-Net level intervention, and a deeper score-based variant, and evaluates them across celebrity, object, and explicit-content erasure. Experiments on multiple erasure methods show that current erasure methods can often be bypassed under this threat model. The paper also discusses how this attack can be used as a stress test for future erasure methods.

**Compliance With Llm Reviewing Policy:**

Affirmed.

**Final Justification:**

The rebuttal does not address my comments very well.  The comparison with [1-3] in the key question Q3 is not presented so that it is hard to evaluate the model.  Thus, I would like to remain my previous opinion.

**Key Questions For Authors:**

1. Can the author have a deeper discussion on the transferability of this pressure test on the DiT model? For example, can EEB_surface be transferred to LLM-based text encoders, can EEB_shallow be transferred to DiT models with Joint-Attention mechanisms, and can EEB_deep be effectively used in larger-scale models?
2. In the black-box setting, has the author considered the impact of the scale of fine-tuning data and toxic data on the attack?
3. The conceptual erasure methods in the paper all rely on text-based erasre, and it seems that the author has ignored a category of work that focuses on erase visual patterns or preference optimization [1][2][3]. Discussing the effectiveness of EEB in them will also be beneficial (Just discuss is fine, no need to supplement with experiments).

[1]SafeGen: Mitigating Unsafe Content Generation in Text-to-Image Models, CCS24

[2]Direct Unlearning Optimization for Robust and Safe Text-to-Image Models, Neurips24

[3]SafetyDPO: Scalable Safety Alignment for Text-to-Image Generation

**Limitations:**

The author did not set up a separate "Limitation" chapter, but discussed the detectability of the method and its potentially misused characteristics in the main text. Other suggestions can be found in the "Weakness" part.

**Strengths And Weaknesses:**

## Strength
1. The paper present a useful threat model of backdoor poisoning under a pre-erasure threat model, and this setting is a meaningful contribution as a stress test of current erasure methods.
2. The experiment is broad, the main claim is supported by experiments across multiple attack variants, three erasure tasks, and several erasure baselines.

## Weakness
1. The author's core idea can be summarized as follows: "If the erasure method relies on a known token and the attacker remaps the concept to another hidden token in advance, the erasure misses this path." This idea is easy to expected that the method does not seem like a complete technical contribution, but rather an empirical test based on existing conclusions.
2. The black-box poisoning results are based on poisoned fine-tuning rather than a more realistic pretraining-scale contamination setting. This is enough to show feasibility, but it does not fully establish how realistic or common this threat is in practice.
3. The experiments are focused mainly on U-Net-based Stable Diffusion models such as SD1.4 and SD2.1. It is necessary to discuss more advanced models, such as DiT-based models. EEB_surface and EEB_shallow do not seem to work directly on the DiT architecture model (due to changes in the text encoder and Joint-Attntion mechanism)

---

> ### Author Rebuttal · Authors · 2026-03-30
>
> We thank the reviewer for recognizing our threat model as a meaningful stress test for concept erasure and for acknowledging the breadth of our experiments.
> # W1 - Expected Vulnerability
> We respectfully disagree that the result is “expected.” That interpretation would be fitting for inference-time prompt filters, where it would indeed be unsurprising that remapping a harmful concept to a token sequence outside the blacklist could bypass the defense. Our paper however studies parameter-level erasure, which aims to remove the target concept from the model’s internals, rather than merely blocking certain prompts. This distinction is already visible in our weakest baseline. Although ESD erases using the single prompt “nudity,” it still suppresses 767 of the 931 sexual prompts in I2P, even though 930 of them do not contain the word “nudity” (e.g. “harry potter in victoria’s secret lingerie.”) Moreover, **we not only evaluate standard erasure methods but also robust ones like RECE, Receler, and AdvUnlearn, which explicitly search beyond the target tokens for embeddings that can still regenerate the erased concept and find that despite elaborate search strategies, these defenses still leave a pre-embedded trigger-target link intact.**
> # W2 - Fine-tuning not Pretraining-scale
> We chose **fine-tuning because it is both tractable and practically relevant**. Nightshade (Shan et al. 2024) reports that a single full training run already requires more than a week of computation. Thus, scaling such experiments to our hundreds of poisoned and erased models is infeasible. As foundation models continue to grow, more **real-world use is driven by continued training rather than training from scratch**. This makes poisoned fine-tuning a realistic attack surface. Our setup also follows prior work that studies attacks through finetuning such as BAGM (Vice et al. 2023)  and TrojVLM (Lyu et al. 2024). Carlini et al. (2024) demonstrated the feasibility of pretraining poisoning, and **Nightshade shows that poisoning T2I models during finetuning can be comparably effective to poisoning during training from scratch**. Together, these findings suggest that our finetuning-based black-box threat model plausibly extends to the pretraining setting.
> # W3 & Q1 - Limited Model Coverage
> Our evaluation focuses on U-Net-based models, because erasure for DiT models is still less mature. That said, we do not view EEB as U-Net-specific. As a targeted backdoor that survives later erasure, EEB transfers naturally across architectures. Only $EEB_{shallow}$ is tied to U-Net-style cross-attention editing, whereas $EEB_{surface}$ and $EEB_{deep}$ transfer naturally to newer architectures. Recent work, Tuning Just Enough (Chen et al. 2026), shows that multi-encoder T2I models can be effectively poisoned via Rickrolling, making the transfer of $EEB_{surface}$ to multi-encoder DiT such as SD3 and FLUX possible. **Thus, we tested two EEB variants on SD3  against object erasure with target `bird`**, applying ESD and EraseFlow (Kusumba et al., 2025) after poisoning:
> |Atk|ESD|EraseFlow|
> |:-|:-:|:-:|
> |$EEB_{surface}$|15.09%|32.58%|
> |$EEB_{deep}$|17.28%|12.52%|
>
> **Notably, $EEB_{surface}$ circumvents the strong EraseFlow erasure on nearly every third prompt on the DiT-based SD3. These preliminary results suggest EEB remains relevant beyond the U-Net setting**.
> # Q2 - Poisoning Rate Sensitivity
> To study this, **we ran an ablation for celebrity erasure under RECE with reduced poisoning ratios**. In the main experiment, we finetuned for 100,000 steps with a 1% poison ratio. For this ablation, we additionally considered 0.1% poisoning:
> |Erasure|Attack|Acc_r|Acc_e|ASR|
> |:-|-|-|-|-|
> |-|-|91.60|92.04|0|
> |RECE|-|70.88|0.12|0|
> |RECE|$EEB_{data}$ (1%)|50.40|9.60|80.16|
> |RECE|$EEB_{data}$ (0.1%)|65.88|4.40|28.48|
>
> As the poisoning ratio decreases, **ASR decreases as well, but remains substantial at 28.48%**. At the same time, less poisoning data also reduces residual target leakage under the erased prompt, reflected in a lower target accuracy ($Acc_e$) from 9.60 to 4.40, making the attack harder to spot for the defender.
> # Q3 - Vision-based Erasure
> This is an astute observation, and we agree that text-agnostic erasure may offer a valid tool against the EEBs proposed in this work. These methods do not rely on textual conditioning to define harmfulness. Instead, they leverage safe/unsafe image pairs to identify harmful noise predictions and suppress those. **The natural analogue of EEB would be a *latent-space EEB*.** Instead of creating only a text-side trigger-concept association, the attacker would associate a trigger with a specific latent-space region that lies outside the latent patterns covered during erasure, so that the sanitized model does not identify it as harmful and continues to predict harmful noise there. Such EEBs would probe whether the text-agnostic erasure objective covers all *latent image regions* through which a backdoor could route harmful generation.

---

> > ### Author Rebuttal · Reviewer_9FP8 · 2026-04-03
> >
> > The responses show the generalizable of the method.  However, in my humble opinion, the authors' responses do not address my major concerns on its assumption and application feasibility.  Please help to clarify it.

---

> > > ### Author Response · Authors · 2026-04-04
> > >
> > > We thank the reviewer for the clarification and for recognizing the generalizability of our threat model. We understand the remaining concern as focusing mainly on **W2** and **Q2**: whether fine-tuning-time poisoning is realistic, and how strongly EEB depends on the poisoning ratio.
> > >
> > > For **Q2**, we expanded the 0.1% poisoning ablation to ESD to corroborate our previous findings that, even in more realistic scenarios with lower poisoning rates, EEB remains a viable threat. Furthermore, we continued fine-tuning the previous 0.1% poisoning checkpoint for an additional 100,000 steps, reaching 200,000 steps in total, to better approximate a full-training regime and test whether the reduced ASR at lower poisoning rates can be recovered through longer training.
> > >
> > > |Erasure|Attack|Acc_r|Acc_e|ASR|
> > > |:-|:-|:-:|:-:|:-:|
> > > |-|-|91.60|92.04|0|
> > > |RECE|-|70.88|0.12|0|
> > > |RECE|$EEB_{data}$ (1%)|50.40|9.60|80.16|
> > > |RECE|$EEB_{data}$ (0.1% & 100,000 steps)|65.88|4.40|28.48|
> > > |RECE|$EEB_{data}$ (0.1% & 200,000 steps)|67.22|4.85|53.16|
> > > |ESD|-|83.88|3.88|0|
> > > |ESD|$EEB_{data}$ (1%)|84.88|5.88|20.80|
> > > |ESD|$EEB_{data}$ (0.1% & 100,000 steps)|84.06|4.91|16.92|
> > > |ESD|$EEB_{data}$ (0.1% & 200,000 steps)|84.76|5.18|19.14|
> > >
> > > The pattern is consistent across methods: lowering the poisoning ratio reduces ASR, but the attack remains effective even at 0.1%, with ASR of 19.14% against ESD. The RECE and ESD results after 200,000 steps of $EEB_{data}$-poisoning further show that **the drop in ASR at lower poisoning rates can be partially recovered through longer training, strengthening the threat in settings closer to full training scale.**
> > >
> > > For **W2**, we believe fine-tuning-time poisoning is practically relevant because many diffusion fine-tuning methods (from various sub-domains, including watermarking and concept erasure itself) rely on prior-preservation regularization to avoid overfitting and mode collapse [1]. This is the case for several methods that regularize on subsets of LAION-400M or other public data [2,3,4]. Since these auxiliary datasets are broader and typically less controlled than the target fine-tuning data, poisoning during continued training is a realistic threat model. While a full pre-training-scale experiment is out of scope for the rebuttal, we refer to our earlier response and to Carlini et al. [5], which shows that pre-training poisoning is practical and effective at scale.
> > >
> > > **Finally, to make EEB a practical community stress test, we will release EEB-poisoned checkpoints so that future erasure methods can be evaluated against them directly.**
> > >
> > > **References**
> > >
> > > [1] Ruiz, N., Li, Y., Jampani, V., Pritch, Y., Rubinstein, M. and Aberman, K., 2023. Dreambooth: Fine tuning text-to-image diffusion models for subject-driven generation. In Proceedings of the IEEE/CVF conference on computer vision and pattern recognition (pp. 22500-22510).
> > >
> > > [2] Kumari, N., Zhang, B., Zhang, R., Shechtman, E. and Zhu, J.Y., 2023. Multi-concept customization of text-to-image diffusion. In Proceedings of the IEEE/CVF conference on computer vision and pattern recognition (pp. 1931-1941).
> > >
> > > [3] Zhang, Y., Yang, M., Zhou, Q. and Wang, Z., 2024. Attention calibration for disentangled text-to-image personalization. In Proceedings of the IEEE/CVF conference on computer vision and pattern recognition (pp. 4764-4774).
> > >
> > > [4] Feng, W., Zhou, W., He, J., Zhang, J., Wei, T., Li, G., Zhang, T., Zhang, W. and Yu, N., 2024. Aqualora: Toward white-box protection for customized stable diffusion models via watermark lora. arXiv preprint arXiv:2405.11135.
> > >
> > > [5] Carlini, N., Jagielski, M., Choquette-Choo, C.A., Paleka, D., Pearce, W., Anderson, H., Terzis, A., Thomas, K. and Tramèr, F., 2024, May. Poisoning web-scale training datasets is practical. In 2024 IEEE Symposium on Security and Privacy (SP) (pp. 407-425). IEEE.

---

### Decision · Program_Chairs · 2026-04-30

**Decision:**

Accept (regular)

**Comment:**

The paper has received three reviews, two of which argue for acceptance (2x A), and one weakly for rejection (WR).

9FP8 is the most critical review (WR) and considers the contribution to be an empirical observation rather than a new method. They consider the paper to be under-evaluated in terms of breadth of models, and poisoning attacks.  They concerns on generalizability were addressed by the rebuttal, but even in the final discussions reiterated that they felt sufficiently strongly on the outstanding issues that they could not recommend acceptance.

nmVP felt the paper was significant in addressing the threat to concept erasure techniques in T2I, but felt the paper lacked contrast against adversarial prompting and did not address certain practicality tradeoffs (stealthiness vs efficacy).  They upgraded their rating to accept (A) post-rebuttal in light of experiments addressing each of these points..

5HWZ had similar concerns on practical detectability and felt the empirical evaluation to be underpowered.  They are convinced by the rebuttal which, as with nmVP, was effective at persuading on the practicality of the approach.

Overall the paper has two strongly supporting reviews, in the sense that all rebuttal concerns were addressed and both reviewers consider the paper important and practical in mitigation of concept erasure attack.  The more negative review is only weakly arguing for rejection, as some but not all the concerns were addressed.  The question is whether the outstanding concerns are severe enough to warrant rejection.  These seem to centre mostly on the baselines provided.  There is some evidence provided in the rebuttal to 5HWZ of performance relative to traditional backdoor attacks in addition to baselines in the paper though it is the case that model breadth is limited.  The AC feels that there is sufficient evidence in support of the technique, and some reasonable baselines included, to demonstrate the validity of the approach.  The majority of reviewers consider the domain contribution important.  Therefore the AC, on balance, considers the paper is acceptable.